# Nitrate-dominated PM$_{2.5}$ and elevation of particle pH observed in urban Beijing during the winter of 2017

Yuning Xie [1], Gehui Wang [1,2][*], Xinpei Wang [1], Jianmin Chen[2,3], Yubao Chen [1], Guiqian Tang [4], Lili Wang [4], Shuangshuang Ge [1], Guoyan Xue [1], Yuesi Wang[4], Jian Gao[5]

[1]Key Laboratory of Geographic Information Science of the Ministry of Education, School of Geographic Sciences, East China Normal University, Shanghai 200241, China

[2]Institute of Eco-Chongming, 3663 N. Zhongshan Rd., Shanghai 200062, China

[3]Department of Environmental Science and Technology, Fudan University, Shanghai, China

[4]State Key Laboratory of Atmospheric Boundary Layer Physics and Atmospheric Chemistry, Institute of Atmospheric Physics, Chinese Academy of Sciences, Beijing 100080, China

[5]Chinese Research Academy of Environmental Sciences, Beijing 100000, China;

*Correspondence to*: Prof. Gehui Wang (ghwang@geo.ecnu.edu.cn)

**Abstrac**t: Chinese government has exerted strict emission controls to mitigate air pollution since 2013, which

resulted in significant decreases in the concentrations of air pollutants such as SO$_2$. Strict pollution control actions also reduced the average PM$_{2.5}$ concentration to a low level of 39.7 μg m$^{-3}$ in urban Beijing during the winter of 2017. To investigate the impact of such changes on the physiochemical properties of atmospheric aerosols in China, we conducted a comprehensive observation focusing on PM$_{2.5}$ in Beijing during the winter of 2017. Compared with the historical record (2014–2017), SO$_2$ decreased to a low level of 3.2 ppbv in the winter of 2017 but NO$_2$ level

was still high (21.4 ppbv in the winter of 2017). Accordingly, contribution of nitrate (23.0 μg m$^{-3}$) to PM$_{2.5}$ far

exceeded over sulfate (13.1 μg m$^{-3}$) during the pollution episodes, resulting in a significant increase of nitrate to sulfate molar ratio. The thermodynamic model (ISORROPIA-II) calculation results showed that during the PM$_{2.5}$ pollution episodes particle pH increased from 4.4 (moderate acidic) to 5.4 (more neutralized) when the molar ratio of nitrate to sulfate increased from 1 to 5, indicating that aerosols were more neutralized as the nitrate content elevated. Controlled variable tests showed that the pH elevation should be attributed to nitrate fraction increase other than crustal ion and ammonia concentration increases. Based on the results of sensitivity tests, the future prediction for the particle acidity change was discussed. We found that nitrate-rich particles in Beijing at low and moderate humid conditions (RH: 20%–50%) can absorb twice amount of water than sulfate-rich particles, and the nitrate and ammonia with higher levels have synergetic effects, rapidly elevating particle pH to merely neutral (above 5.6). As moderate haze events might occur more frequently under the abundant ammonia and nitrate-dominated PM$_{2.5}$ conditions, the major chemical processes during haze events and the control target should be re-evaluated to obtain the most effective control strategy.

## 1 Introduction

Severe haze pollution has been causing serious environmental problems and harming public health in China over the past decades (He et al. 2001; Wang et al. 2016; Zhang et al. 2015b). Therefore, strong actions have been taken to improve the worsening atmospheric environment, including cutting down the pollutant emissions with forced installation of catalytic converter on vehicles, building clean-coal power generation system, prohibiting open burning of crop residue during the harvest seasons, etc. (Chen et al. 2017; Zhang et al. 2012; Liu et al. 2016). As a result, the PM$_{2.5}$ pollution occurrence is reduced to meet the goals in Air Pollution Prevention and Control Action Plain (issued by the State Council of China, http://www.gov.cn/zwgk/2013-09/12/content_2486773.htm, in Chinese). Among all the regions of interests, Beijing has achieved great success in PM$_{2.5}$ reduction (the annual average PM$_{2.5}$ concentration of 2017 was 58 μg m$^{-3}$). Yet, PM$_{2.5}$ concentration in Beijing is still higher than that in most developed countries.

There are many factors contributing to the PM$_{2.5}$ pollution in China (Guo et al. 2014; Ding et al. 2013). The PM$_{2.5}$ pollution across the country is featured by significantly high secondary formation of inorganic components (Huang et al. 2014a). Sulfate, nitrate and ammonium (SNA) comprised over 30% of the PM$_{2.5}$ mass, and SNA's fraction continues to increase during the pollution evolution (Cao et al. 2012). While models could well predict the airborne particle pollution in the U.S. or Europe, it is challenging to simulate the real atmospheric pollution in China (Wang et al. 2014; Ervens et al. 2003). Previous modeling works showed that the simulated PM$_{2.5}$ concentrations were underestimated within the current scheme, which is related to the important role of heterogeneous reactions in the SNA formation processes (Huang et al. 2014b; Herrmann et al. 2005). It was reported that the classical formation mechanism of sulfate in the atmosphere was through oxidation by H$_2$O$_2$. But recent studies in China pointed out that non-classical formation pathways cannot be ignored. In Beijing, severe haze events occur with abundant nitrogen species (NO$_x$, NH$_3$, etc.), high relative humidity (RH) and less active photochemistry (Wang et al. 2016; Cheng et al. 2016). Field observations, chamber experiments, source apportionments and numerical simulation works all suggest that the joint effect of NO$_2$, SO$_2$, and NH$_3$ is important for the sulfate formation processes in haze events (Cheng et al. 2016; Wang et al. 2016; Wang et al., 2018; He et al. 2018, Xue et al., 2019). Aqueous oxidation of SO$_2$ by NO$_2$ could be a major process of sulfate formation in Beijing during winter, as well as the catalyzed oxidation by transition metal ions (TMI) (Wang et al. 2016; Cheng et al. 2016). Besides, although the photochemistry is less active during haze periods in winter, the extra OH radical provided by HONO might enhance the atmospheric oxidation capacity and lead to a rapid formation of SNA (Tan et al. 2018, Ge et al. 2019). Since these reactions are all sensitive to particle acidity, adequate quantification of airborne particles' acidity is essential for elucidating the specific contribution.

Particle acidity has been widely studied due to its important role in the haze formation, and has been widely implemented in major models (Yu et al. 2005; Oleniacz et al. 2016). Since the practical method of directly measuring the particle acidity in real atmosphere is not available (Wei et al. 2018; Freedman et al. 2019), thermodynamic models have been mostly used in quantifying the particle acidity. Most models (ISORROPIA II, E-AIM-IV,

AIOMFAC, etc.) can predict $H^+$, aerosol liquid water content (ALWC) and the partitioning of volatile/semi volatile components, such as ammonia (Fountoukis and Nenes 2007; Clegg et al. 2008). These models' abilities to describe

physiochemical properties of airborne particles have been validated in previous studies (Weber et al. 2016; Guo et al. 2016; Shi et al. 2017; Tao and Murphy 2019; Murphy et al. 2017). However, several publications using the same method gave different particle pH values in Beijing, and contradictory conclusions were drawn on the importance of the sulfate formation by $NO_2$ oxidation. Cheng et al. (2016) conducted some modeling work and suggested that the $PM_{2.5}$ pH in Beijing ranges between 5.4 and 6.2, which is favorable for the aqueous $NO_2$ oxidation. The $NO_2$

oxidation's major contribution and the importance of high ALWC and sufficient ammonia are supported not only by modeling works, but also by field observation and chamber study (Wang et al. 2016; Chen et al. 2019). On the contrary, Liu et al. (2017) simulated the particle pH during the winter of 2015 and 2016 with the same method, and claimed that pH of the Beijing haze particles was lower (3.0–4.9, average 4.2) and unfavorable for the $NO_2$ oxidation mechanism. Based on the ISORROPIA-II model results , which assumed Chinese haze particles as a

homogeneous inorganic mixture, Guo et al. (2017) further concluded that high ammonia cannot raise the particle pH enough for the aqueous oxidation of $SO_2$ by $NO_2$. Recently Song et al. (2018) reported that the thermodynamic model (ISORROPIA-II) has coding errors, which can lead to the predicted pH values negative or above 7. Furthermore, with lab studies and field observations, Wang et al. (2018) raised the concern that whether it is appropriate to elucidate sulfate production for the Beijing haze by using particle pH predicted only based on the

inorganic compositions. In fact, since the real atmosphere is affected by uncountable factors, it is common that particle pH has variation when simulated with the ambient data. Although the pH predicted by the thermodynamic models are of uncertainty, it is widely believed that haze particles in China are moderate acidic and are more neutralized than those in the US., given that the gaseous ammonia is still at a high level relative to particulate ammonium(Song et al. 2018).

Air pollution control in China has entered the second phase—further mitigation of the moderate haze pollution, which is characterized by high levels of nitrate and ammonium and low level of sulfate (Liu et al. 2019; de Foy et al.

2016) due to the efficient $SO_2$ emission control. Such a change in chemical compositions could significantly alter physicochemical properties of the atmospheric aerosols in China. This paper aims to investigate the variation of the particle acidity of $PM_{2.5}$ as sulfur emission is well controlled and nitrogen oxides emission remains high in Beijing.

First, the compositions of air pollutants, including inorganic components of $PM_{2.5}$ during the winter of 2017 and 2018, were analyzed and compared with previous studies; then, based on observations, the response of particle acidity to the elevation of nitrate was studied by using the ISORROPIA II thermodynamic model; finally, the possible changes in the future are discussed based on the sensitivity tests.

**2 Sampling site and instrumentation**

The observation was conducted at an urban site—the State Key Laboratory of Atmospheric Boundary Layer Physics and Atmospheric Chemistry, Institute of Atmospheric Physics (IAP), Chinese Academy of Sciences (39°58′28″N, 116°22′16″E) in Beijing. All the instruments were on the roof of a two-story building. The local emissions are mainly from the vehicles, and the industrial emission is greatly reduced since the major factory/power plants are moved out of Beijing or phased out due to the emission control policy. Overall, this site represents the

typical atmospheric environment in urban Beijing, from which the data obtained can be compared with those from previous studies in the city (Ji et al. 2018).

    A continuous online measurement of atmospheric components was conducted with a time resolution of 1 hour. Two TEOM$^{TM}$ continuous ambient PM monitors using $PM_{2.5}$ or $PM_{10}$ cyclone inlet (Metone) were applied to obtain $PM_{2.5}$ and $PM_{10}$ mass concentrations. For trace gases ($O_3$, $NO_2$ and $SO_2$), a series of gas monitors were used for the

hourly measurement (Model 49i, 42i and 43i, respectively). Meteorology data, including ambient temperature, RH, wind speed, wind direction and total solar radiation, were measured with an automatic weather station (MILOS 520, VAISALA Inc., Finland) located in the middle of the observation site yard. Visibility data of Beijing were downloaded from the open database (https://gis.ncdc.noaa.gov/maps/ncei/cdo/hourly). Apart from these online monitors, a high-volume sampler (TISCH ENVIRONMENTAL) with a $PM_{2.5}$ inlet was used to collect $PM_{2.5}$

samples on a day/night basis (daytime of 8:00–17:50 and nighttime of 18:00–7:50).

The inorganic water-soluble components of $PM_{2.5}$ ($SO_4^{2-}$, $NO_3^-$, $Cl^-$, $NH_4^+$, $Na^+$, $Ca^{2+}$, $Mg^{2+}$ and $K^+$) and ammonia gas were measured with an online-IC system: IGAC (In-situ Gases and Aerosol Composition monitor, Fortelice International Co., Ltd.). IGAC comprises of 2 parts: sampling unit and analyzer unit (Young et al. 2016). A vertical wet annular denuder (WAD) is used to collect the gas-phase species prior to a scrub and impactor aerosol

collector (SIC), which can efficiently collect particles into liquid samples. During the campaign, 1mM $H_2O_2$ solution is used as the absorption liquid for the air samples. Under most atmospheric conditions, the absorption liquid can efficiently absorb the target atmospheric components (e.g. $SO_2$). An ICS-5000$^+$ ion chromatograph is used as the analyzer unit in this study. For anions, an AS18 column (2 mm $\times$ 250 mm, Dionex™ IonPac™) is used while a CS-16 column (4 mm $\times$ 250 mm, Dionex™ IonPac™) is chosen to analyze major cations, both running with

recommended eluent (solution of KOH for anion/methane sulfonic acid for cation). The performance of the IGAC system has been tested and improved over recent years, and studies of $PM_{2.5}$ water-soluble ion observations have been conducted by using it (Young et al. 2016; Song et al. 2018; Liu et al. 2017). A better sensitivity due to the advanced suppression technology of the system greatly enhances its ability to measure trace ions, such as sodium and magnesium, which is important in studies of particle ion balance. For details of the comparison between IGAC

and filter sampling results, please refer to supplementary materials (Fig. S1).

## 3. Results

### 3.1 Major pollutants' levels

We firstly present the overall time series and statistics of major pollutants' concentration and meteorological parameters from Dec. 15$^{th}$, 2017 to Feb. 25$^{th}$, 2018. As shown in Fig.1, during the observation campaign, Beijing

was relatively cold and dry. Due to the frequent cold-air outbreaks, the average air temperature was around 0°C with a minimum of −10°C, and the RH was low on average (20%–30%) with a maximum of 80%. The average total solar radiation was 254.3 w m$^{-2}$, which is typical in Beijing during winter. The wind usually blew from the north with an

average speed of 1.9 m s$^{-1}$, but the strong wind (over 5 m s$^{-1}$) frequently occurred on the clean days. Benefiting from the weather condition, the atmospheric pollution in Beijing was much weaker than that in the winter of 2013. Overall, the improvement of the atmospheric environment was visible: the average visibility was around 15 km during the campaign and about 7.5 km during the pollution periods.

With strict control actions, there were less PM$_{2.5}$ pollution episodes and its concentration kept at a low level during most of the time in the winter of 2017. The average concentrations of PM$_{2.5}$ and PM$_{10}$ were 39.7 μg m$^{-3}$ and 68.5 μg m$^{-3}$, respectively. According to the PM$_{2.5}$ concentration, three conditions of the atmospheric environment were classified in this study : clean (the PM$_{2.5}$ was below 35 μg m$^{-3}$), transition (the PM$_{2.5}$ was between about 35 μg m$^{-3}$ and 75 μg m$^{-3}$) and pollution (the PM$_{2.5}$ was above 75 μg m$^{-3}$). In the clean, transition and pollution periods, the average PM$_{2.5}$ concentrations were 13.0±7.8 μg m$^{-3}$, 52.0±11.4 μg m$^{-3}$ and 128.0±46.5 μg m$^{-3}$, respectively (as showed in Table 1), indicating that there was still PM$_{2.5}$ pollution (maximum hourly concentration 298 μg m$^{-3}$) during the winter. The average ozone concentration was 18.5±12.8 ppbv, and its value decreased as PM$_{2.5}$ concentrations increased. The average SO$_2$ concentration (3.2±3.1 ppbv) was almost 10 times lower than that of NO$_2$ (21.4±14.8 ppbv). This significant contrast between SO$_2$ and NO$_2$ concentrations can be attributed to the sulfur emission control over recent years and the fast increase of gasoline vehicles in Beijing (Cheng et al. 2018; Wang et al. 2018b). All gaseous pollutants showed an increasing trend as the PM$_{2.5}$ concentration increased during the haze episodes, while NO$_2$ concentration elevation was the largest .

NO$_2$ and SO$_2$ are the most important precursor gases for inorganic nitrate and sulfate in PM$_{2.5}$. Sulfur emission control drastically reduced the ambient SO$_2$ concentration while NO$_2$ lacked effective control policy. To better describe this situation, changes of these two precursor gases during winter are investigated by examining the data from 2014 to 2016 in Beijing. Average values and the standard deviation are plotted in Fig. 2. SO$_2$ showed a significant decreasing trend in all the three conditions. In 2014, SO$_2$ concentration were 3.9, 10.0 and 16.9 ppbv at clean, transition and pollution periods, respectively. The SO$_2$ concentration difference in different pollution levels was narrowing. Until 2017, the difference of SO$_2$ concentrations between any two of three conditions had been all

within 10 ppbv. Meanwhile, $NO_2$ concentrations kept increasing after 2015 in clean and transition conditions, but the $NO_2$ concentration during the pollution periods of 2017 was unexpectedly lower than 2014 records. This significant drop of $NO_2$ concentration in $PM_{2.5}$ pollution proves the effectiveness of pollution control in 2017, such as

construction prohibition, private vehicle restriction and LNG promotions in neighboring regions (Cheng et al. 2018).

**3.2 $PM_{2.5}$ chemical compositions**

Comparing to several previous reports, the chemical compositions of $PM_{2.5}$ during the winter of 2017 in Beijing changed significantly (Shao et al. 2018; Elser et al. 2016; Ge et al. 2017; Huang et al. 2017; Wang et al. 2017). The major inorganic ions of $PM_{2.5}$ in Beijing during the winter of 2017 included ammonium ($3.3\pm4.4$ µg m$^{-3}$), nitrate

($7.1\pm9.6$ µg m$^{-3}$), sulfate ($4.5\pm5.9$ µg m$^{-3}$) and chloride ($2.4\pm2.3$ µg m$^{-3}$). Concentrations of the major components increased as the $PM_{2.5}$ concentration increased, but changes in the crustal ion ($Na^+$, $Mg^{2+}$ and $Ca^{2+}$) concentrations were less significant. $K^+$ increased during the $PM_{2.5}$ pollution episodes (average concentration: $2.3\pm5.1$ µg m$^{-3}$), indicating the possible contribution of biomass burning sources or fireworks during Chinese New Year. $Cl^-$ in $PM_{2.5}$ has been used as a tracer for biomass burning and the coal consumption. The concentration of $Cl^-$ (average

concentration $2.4\pm2.3$ µg m$^{-3}$) increased significantly as $PM_{2.5}$ increased, but the imbalance of chloride molar concentration to potassium (average $K^+$:0.059 µmol m$^{-3}$ vs. average $Cl^-$ 0.13 µmol m$^{-3}$) suggests that biomass burning might not be the major source of $PM_{2.5}$ chloride other than the coal consumption during the $PM_{2.5}$ pollution episodes in Beijing.

Concentration of sulfate, nitrate and ammonium greatly increased the $PM_{2.5}$ pollution. Different from previous

records (Wang et al. 2016; Huang et al. 2014a; Ji et al. 2014), nitrate dominated the water-soluble ions (WSIs) in the winter of 2017. During pollution episodes, concentration of nitrate and sulfate were $23.0\pm10.7$ µg m$^{-3}$ and $13.1\pm8.4$ µg m$^{-3}$, with an average molar ratio of nitrate to sulfate (Ratio $_{\text{N-to-S}}$) around $3.3\pm1.4$. Fig.3 shows the scatterplot of nitrate and sulfate with the total water-soluble ions. Sulfate comprised a lower fraction when total WSIs was below 65 µg m$^{-3}$, but the fraction increases as WSIs exceeds 65 µg m$^{-3}$, showing an enhanced formation of sulfate during

heavy pollution episodes. Interestingly, the ratio of nitrate to WSIs remained the same throughout the campaign. As the concentrations of other components also increased, this phenomenon indicated that the nitrate formation was enhanced in hazy days. Besides, the concentrations of ammonium and ammonia both increased significantly (ammonium: $0.9\ \mu g\ m^{-3}$ to $10.4\ \mu g\ m^{-3}$; ammonia: $4.3\ \mu g\ m^{-3}$ to $12.9\ \mu g\ m^{-3}$) from clean to pollution conditions.

### 3.3 Comparison of major inorganic compositions during the early 21$^{st}$ century in Beijing

To illustrate the changes in chemical compositions of PM$_{2.5}$ at the China's economy booming stage (1999–2017), nitrate, sulfate and ammonium are chosen for the comparison with previously reported data during winter in Beijing (Fig. 4). Only winter-averaged observation data or representative pollution records are selected for the illustration on changes of SIA compositions. On average, although the concentration might have been varied due to different emissions and weather conditions over the years, SIA concentration in the winter of 2017 was the lowest compared with the years before. Sulfate concentration varied from $4.5\ \mu g\ m^{-3}$ to $25.4\ \mu g\ m^{-3}$ and contributed the most PM$_{2.5}$ masses among SIA species during the pollution episodes before 2015. The emission control of SO$_2$ started in 2006 to prevent adverse atmospheric environment events such as acid rain and high particulate matter loading (Wang et al. 2013; Wang et al. 2018b). As a result, the sulfate concentration in winter decreased gradually (see results of 1999, 2011, 2015-a and 2017) until recent years' record has been much lower than that in the early 2000s (detailed literature comparison can be found in Lang et al. (2017)). However, it was widely reported that sulfate still contributed the most to the PM$_{2.5}$ mass concentration during the severe haze periods, such as the winter of 2013 (Huang et al. 2014a; Guo et al. 2014; Ji et al. 2014). The heterogeneous formation might be responsible for the enhanced conversion ratio from SO$_2$ to particulate sulfate, including the NO$_2$-promoted aqueous reaction and transition-metal-catalyzed oxidations (Huang et al. 2014b; Xie et al. 2015). On the other hand, the NO$_x$ emission in North China significantly increased as the power consumption and vehicle amount kept increasing. Therefore, nitrate in PM$_{2.5}$ had been increasing since 2011. The average concentration of nitrate rose from $7.1\ \mu g\ m^{-3}$ to $29.1\ \mu g\ m^{-3}$. By 2015, the nitrate concentration had exceeded the sulfate concentration, and both compositions contributed

equally to PM mass in winter pollution episodes. Although the nitrate concentration during pollution periods decreased in 2017 (23.0 μg m$^{-3}$), the decrease was not significant and the concentration was still comparable to

previous studies. The winter-averaged ammonium concentration reached the maximum (~20 μg m$^{-3}$) in 2015, but decreased afterwards. In a word, as the dominant composition, high nitrate fraction has become one of the major features of PM$_{2.5}$ in Beijing during winter.

The ratio between major SIA components can better represent the composition change discussed above. As shown in Fig.5, the nitrate to sulfate ratio (Ratio$_{N-to-S}$) has been increasing significantly from below 1.0 to 2.7 (1999

vs. 2017). Ratio$_{N-to-S}$ was around 1 before 2013 but then steadily increased after 2013, same as previous publications (Shao et al. 2018; Lang et al. 2017). Interestingly, Ratio$_{N-to-S}$ during pollution episodes was lower than the winter average value in 2015, but Ratio$_{N-to-S}$ during pollution episodes greatly exceeded the average value in 2017, showing the dominance of nitrate in the PM$_{2.5}$ pollution. The rapid increase of Ratio$_{N-to-S}$ not only resulted from the sulfur emission control, but also from more nitrate partitioning to the particle phase. Abundant ammonia in Beijing's

atmosphere can enhance the partitioning of nitric acid gas by forming ammonium nitrate. To identify whether the ammonia is sufficient, the ammonium to sulfate ratio (Ratio$_{A-to-S}$) is calculated with the published data as well. It is reported that North China Plain experienced ammonia insufficiency during summer (Ratio$_{A-to-S:}$ less than 1.5), limiting the formation and partitioning of nitrate into the particle phase (Pathak et al,. 2004; Pathak et al. 2009; Pathak et al. 2011). However, Ratio$_{A-to-S}$ in Beijing during winter was always above 1.5. The lowest value appeared

in 1999 (averaged Ratio$_{A-to-S}$: 1.7), then the ratio increased rapidly (above 3) after 2011 (red bars in Fig.5). In recent years, the Ratio$_{A-to-S}$ has reached over 4. This value is typically observed in the eastern U.S during winter, though the absolute concentration is much higher in Beijing (Shah et al. 2018). To sum up, the effective sulfur emission control and ammonia-rich atmosphere provide the favorable environment for nitrate formation, and eventually change PM$_{2.5}$ in Beijing from sulfate-dominated to nitrate-dominated type.

**3.4 Aerosol pH's response to the elevation of nitrate fraction in PM$_{2.5}$**

The shift from sulfate-dominated to nitrate-dominated PM$_{2.5}$ further influences the secondary chemical processes via changing physiochemical properties of aerosols, e.g. hygroscopicity and particle acidity. In a thorough study in the U.S, despite the well control of NO$_x$ emission, the nitrate fraction in PM$_{2.5}$ didn't show a corresponding decreasing trend. It was caused by the elevated partitioning of nitric acid to the particle phase in the eastern America
(Shah et al. 2018). Researchers implied that higher nitrate partition fraction was resulted from the increasing particle pH, while some studies showed that the particle pH was decreasing as particulate sulfate decreases in the U.S (Weber et al. 2016). To better understand the correlation between particle pH and chemical compositions, it is necessary to engage simulations with high-resolution observation datasets which covers as much pollution types as possible.

In this study, the bulk particle pH is calculated with the thermodynamic model ISORROPIA II in forward mode
with the assumption of aerosol in metastable state. The simulation is limited to the data with the corresponding RH between 20%–90%, same as that in previous studies (Liu et al. 2017; Cheng et al. 2016). The analysis is further limited to data with sufficient ALWC (above 5 μg m$^{-3}$) to avoid unrealistic pH values caused by false predictions of ALWC. To study the effect of the nitrate fraction's elevation on particle acidity, the Ratio$_{N-to-S}$ is compared to the bulk particle pH (Fig. 6). As the nitrate fraction increases, the particle pH increases. When the Ratio$_{N-to-S}$ is between
0–2, predicted pH values are rather scattered (2.1~6.2) with a median value of 4.4. As the ratio increases, pH values become less scattered and the median value increases as well. When the ratio is around 4–6, the predicted pH values range from 4.9 to 5.6 with a median value of 5.4, which is comparable with previous reported values in PM$_{2.5}$ pollution episodes (Cheng et al. 2016; Wang et al. 2016; Xie et al. 2015). There are several possible explanations that could lead to pH increasing with higher Ratio$_{N-to-S}$, including neutralization by ammonia, higher pH of
ammonium nitrate in comparison with ammonium sulfate, and increased ALWC leading to dilution of predicted H$^+$ (Hodas et al. 2014; Xue et al. 2014; Wang et al. 2018c). To confirm that the pH elevation is not caused by crustal ions, the simulation using data without crustal ions (input is set to 0) was conducted. It is shown that the exclusion of crustal ions in the simulation can cause an overall lower pH, but the pH elevation with Ratio$_{N-to-S}$ is still

observed (detailed analysis can be found in the supplementary materials, Fig.S2 and S3). On the other hand, as a major controlling factor (Guo et al. 2017; Song et al. 2018), ammonia concentration was even lower at the nitrate dominated condition (Fig.S4). In order to further elucidate the relationship between nitrate fraction and pH, a controlled sensitivity test was conducted for the Beijing $PM_{2.5}$ aerosols by replacing particulate sulfate with the same moles of nitrate and keeping all other variables constant such as RH, temperature, ammonia, anions and cations. As shown in Fig.7, the median values of simulated pH increased from 4.6 to 5.1 as the transferring fraction increases from 10% to 80%. When the fraction exceeded 80%, pH median value was a bit lower compare to the $pH_{80\%}$ (Fig.7a). By examining the difference between the sensitivity test results and the original pH values, an overall increase of pH by ~0.5 was found when the fraction exceeds 60% (Fig.7.b). The detailed sensitivity results and description also showed that the pH elevation due to the replacement of sulfate by nitrate was observed at all conditions (Fig.S5). With the fact that other variables remained controlled, the test results reconfirm that as nitrate become dominant, particle pH would significantly increase.

In this study, fewer predicted $H^+$ ions in aerosol liquid water were found to be the major cause of the higher pH with high nitrate fraction. The correlation between $H^+$ and major anions ($HSO_4^-$, $SO_4^{2-}$, $NO_3^-$, $Cl^-$) is shown in Fig.8 to identify the acidity contribution of each anion. Sulfate and bisulfate have long been recognized as major acidic components of atmospheric particles. Their concentrations have significant impacts on the particle acidity (Weber et al. 2016; Liu et al. 2017). Therefore, the $H^+$ ion concentration was found to strongly correlate with sulfate as well as bisulfate (Figs. 8a and 8b). The outlier data points can be attributed to the fireworks events during the Chinese New Year (extreme data on Chinese New Year's Eve are excluded). The average molar ratio of bisulfate to sulfate is 1.08 $\times$ $10^{-6}$, indicating that most of the sulfate is balanced by ammonium, same as the results reported by previous studies (Song et al. 2018). The excess ammonium is then balanced by nitrate and chloride. The correlation between $H^+$ and nitrate ion is much different as ALWC varies (Fig. 8c). Under the high-ALWC condition, the $H^+$ increases with the nitrate concentration, which can be explained by the simultaneously increasing sulfate fraction during several pollution episodes. Under the drier condition (ALWC < 10µg m$^{-3}$), as $NO_3^-$ increases, $H^+$ decreases, which

implies that the weaker aerosol acidity favors nitric acid partitioning to the particle phase. Since HCl is more volatile than nitric acid gas, its occurrence in the particle phase is more sensitive to the particle acidity (Fig. 8d). Therefore, the negative correlation with $H^+$ is much obvious when it comes to chloride, free of ALWC amount.

The low level of $H^+$, especially its negative correlation with Ratio$_{N-to-S}$ should be attributed to the neutralization by ammonia via gas-particle partitioning. Under most conditions, the excess of ammonia is an implicit prerequisite for SIA formation in Beijing, and higher $NH_3$ concentration could increase the predicted particle pH (Guo et al. 2017; Weber et al. 2016). As an auxiliary evidence, ammonia partition fraction ($F_{NH4}$, calculated with observation data) exhibits a positive trend as Ratio$_{N-to-S}$ increases (Fig. 9), while ammonia concentration remains less varied in the same case (Fig.S4). The positive trend is divided into two parts: a more acidic (pH below 4.5) branch with Ratio$_{N-to-S}$ of 1–3 and $F_{NH4}$ of 0.1–0.6 and the other less acidic (pH above 5.5) branch with Ratio $_{N-to-S}$ of 1–7 and $F_{NH4}$ of 0.1–0.4. The overall higher $F_{NH4}$ in the lower pH branch is reasonable since it is more favorable for ammonia partitioning to the particle phase when airborne particles exhibit higher acidity. Moreover, sulfate can accommodate twice amount of ammonia than nitrate and thus increase $F_{NH4}$. Yet, the highest value of $F_{NH4}$ were observed with more nitrate (Ratio $_{N-to-S}$ ~2.5). By contrast, even though the high particle pH (5~6) may suppress ammonia from partitioning to the particle phase (Guo et al. 2017), elevation of $F_{NH4}$ with increasing Ratio $_{N-to-S}$ (1–4) is still observed with pH ranging 5 to 6 despite there were some outliers with lower $F_{NH4}$ and lower $NH_3$ concentration. Nitrate formation is observed to be enhanced in North China, either by heterogeneous formation (e.g. $N_2O_5$ hydrolysis) or with sufficient ambient $NH_3$ (Wang et al. 2013). The positive trend of $F_{NH4}$ with Ratio$_{N-to-S}$ clearly shows that nitrate formation and partitioning have significant contribution to $NH_3$ partitioning process, and will lead to an enhanced neutralization with the help of more ammonia partitioning into the particle phase. Combining these analyses, we conclude that the increasing nitrate fraction in fine particles will lead to higher particle pH in Beijing during winter.

## 4. Discussion: possible impacts of increasing fraction of nitrate in PM$_{2.5}$

So far, the effect of emission control on SNA compositions in Beijing's PM$_{2.5}$ pollution and the response of particle pH have been illustrated, but it is important to make predictions with the current knowledge, providing scientific evidence for future better control strategy. In this section, sensitivity tests regarding to hygroscopicity and particle-acidity change are conducted to help understand the possible changes of these properties in the future.

ALWC is directly engaged in the calculation of particle pH and limited by several major parameters (RH, hydrophilic composition concentration, temperature, etc.). During the campaign, the ALWC predicted by ISORROPIA II varied between 0.8 and 154.2 μg m$^{-3}$ with an average value of 6.4 μg m$^{-3}$. As previously mentioned, the average value of Ratio$_{N-to-S}$ is around 2 during the haze events in the winter of 2017 in Beijing. The average ALWC in the haze events increased to 24.4 μg m$^{-3}$ accordingly. It has been reported that nitrate salts have greater contribution to ALWC due to its lower deliquescence RH, and the elevated ALWC might has strong impact on water-soluble gas partitioning such as glyoxal, leading to an enhanced SOA production (Hodas et al. 2014; Xue et al. 2014). As a matter of fact, the increase of hygroscopicity related to nitrate-rich fine particles has been observed in Beijing (Wang et al. 2018c). However, it is difficult to conclude that lower pH in nitrate-rich particles is caused by the dilution of H$^+$ with higher ALWC with current data, since the higher nitrate fraction is usually observed with the moderate RH in pollution episodes in the winter of 2017. Furthermore, the elevation of pH due to Ratio$_{N-to-S}$ increase is one unit of pH, which means the ALWC shall increase 10 times of amount. This assumption is not supported by current data, since ALWC remain less varied as Ratio$_{N-to-S}$ increases (Fig.S6). Similar with ALWC, the correlation between ambient temperature and Ratio$_{N-to-S}$ is low, further proving that the increase of pH is not caused by change of thermodynamic state (Fig.S7).

The possible enhancement of hygroscopicity in nitrate-rich PM$_{2.5}$ was investigated. Most single salts can only be deliquesced over a certain RH, thus the ALWC only exists when a certain RH is exceeded (Wexler and Seinfeld 1991; Mauer and Taylor 2010). In real atmosphere, aerosols are usually a mixture of salts and organics, which might be easier to deliquesce. In addition, the deliquescence of NH$_4$NO$_3$ is unique, and it becomes more complicated in the

system of $NH_4NO_3/(NH_4)_2SO_4$. The $NH_4NO_3$ system would absorb water vapor even at an extreme low RH, e.g. down to 10% (Willeke et al. 1980; ten Brink and Veefkind 1995). Previous studies show that the system comprised of $NH_4NO_3/(NH_4)_2SO_4$ has a higher deliquesce point when the sulfate content is higher ($Ratio_{N-to-S}<1$), but absorbs water even at low RH (~20%) when nitrate is dominant ($Ratio_{N-to-S} > 2$) in $PM_{2.5}$ inorganic ions (Wexler and Seinfeld 1991; Ge et al. 1996, 1998). Inspired by these facts, we conducted a sensitivity test of ALWC to RH by using the observation dataset to study the effect of nitrate fraction elevation on ALWC changes (the RH value ranging from 20% to 90%, with 10% as interval). Concentrations of pollutants in the clean periods are relatively low and the data of the clean periods might be more influenced by the observation artifacts. Thus, only the data obtained in the transition and pollution conditions were analyzed here. The ALWC changes are defined as Eq (1).

$$Fraction_{change} = ALWC_{(RH+10\%)}/ALWC_{RH} \tag{1}$$

Then, we choose the data with $Ratio_{N-to-S}$ above 3 and $Ratio_{N-to-S}$ below 1. These values were both mentioned in previous lab studies (Ge et al. 1998) and are also typical values of nitrate-rich or sulfate-rich conditions in field observations. As shown in Fig. 10, $PM_{2.5}$ with higher $Ratio_{N-to-S}$ adsorbs more water than lower nitrate fraction as the RH increases, which is more significant under lower RH (<50%) conditions compared with that uner higher RH (50%–70%) conditions. As the RH is usually lower (30%–50) at the beginning stage of $PM_{2.5}$ pollution development in Beijing, such a significant increase in hygroscopicity of nitrate-rich particles can greatly promote the haze formation under relatively dry conditions by enhancing the gas-to-particle partitioning of water-soluble compounds and the aqueous-phase formation of secondary aerosols, e.g. ammonia partitioning and nitrate formation through partitioning or hydrolysis of $N_2O_5$ (Badger et al. 2006; Bertram and Thornton 2009; Sun et al. 2018; Hodas et al. 2014; Shi et al. 2019).

The response of particle pH to ammonia and sulfate changes has been studied in previous studies (Weber et al. 2016; Guo et al. 2017; Murphy et al. 2017). Here we further analyze the particle pH under the elevated nitrate concentration with the increasing ammonia in the atmosphere, which is the possible situation for most Chinese

cities in the coming years. Two kinds of pH sensitivity tests are conducted: one with fixed nitrate but varying sulfate and ammonia and the other with fixed sulfate input but varying nitrate and ammonia (Fig. 11). In the test, crustal ions were all set as 0, while fixed chloride, sulfate and nitrate concentrations were set as the average data in pollution (see Table 2). Compared with previous studies (Guo et al. 2017; Song et al. 2018), the RH was set as 58% and the temperature was set as 273.15K. Despite system errors due to the instability of the model at the extreme high-anion and low-$NH_x$ condition (Song et al. 2018), the pH shows continuously changing as the free variable changes. The significant sharp edge of pH values in both plots defines the ion balance condition. We selected the observations data obtained during the pollution episodes within the RH ranging from 50% to 70% to compare with the results of both sensitivity tests. As shown in Fig.11, apart from some data points (those with lower nitrate concentration but very high $NH_x$ concentration), observation data (triangle points) are quite well merged into the results of sensitivity tests, and the pH values are generally higher than the test results due to the lack of crustal ions inputting the sensitivity simulation. Therefore, the result of sensitivity tests can well represent the pH change of the real atmosphere environment in Beijing.

Future changes in particle pH can be found with the sensitivity test results. Cutting down the sulfate concentration without reducing atmospheric ammonia (horizontally moving from right to left in Fig. 10, left part) can lead to a significant increase of particle pH (up to 5). As can be seen from the right part of Fig.10, the elevation of particle pH might be enhanced with the help of more nitrate in $PM_{2.5}$. The effect of nitrate on particle pH greatly relies on the $NH_x$ concentration in the atmosphere. As the ammonia in the atmosphere over North China might still be increasing (Liu et al. 2018), and the sulfur content in the atmosphere might not be greatly reduced in the future, the particle pH shall increase in the path along the ion balance edge, which also implies a synergetic effect of increased nitrate and ammonia.

These results (lower acidity, higher hygroscopicity) provide insights into the effects of an elevated nitrate content on the physiochemical properties of particles. First, heterogeneous reactions that don't need high acidity might greatly contribute to the airborne particle chemistry, such as the $NO_2$-induced oxidation of $SO_2$ mechanism

(Cheng et al. 2016; Wang et al. 2016). Reactions which rely greatly on acidified particles might contribute less, such as the acid-catalyzed SOA formation from VOCs (Jang et al. 2002; Surratt et al. 2010). Second, the uptake processes of gaseous compounds onto particles (carbonyl acids, for example) might be enhanced, and the uptake of alkaline compounds could also be enhanced via the ALWC elevation. Third, optical properties of particles will greatly vary.

On one hand, higher ALWC can increase the light scattering effect (Titos et al. 2014), while the light absorption by BrC would be enhanced at higher pH (Phillips et al. 2017). All these facts might add up the difficulties to the control of moderate haze in Beijing, which usually occurs with lower RH and higher nitrate content as shown in this study. It is strongly suggested that the control strategy should be made accordingly based on thorough and scientific evaluation on both $NO_x$ and ammonia.

**5 Conclusions**

Due to the strict emission controls, $PM_{2.5}$ in Beijing during the winter of 2017 greatly decreased to a low level ($39.7$ μg m$^{-3}$ for average concentration), but moderate haze episodes still frequently occurred in the city. With the observation and historical data, we found that the $SO_2$ concentration decreased significantly while the $NO_2$ concentration far exceeded that of $SO_2$ and kept increasing in Beijing during winter. In response to the emission

control, the nitrate concentration exceeded the concentration of sulfate significantly and thus became the dominant SIA component in fine particles. The molar ratio of nitrate to sulfate kept increasing over the years and rose to 2.7 during $PM_{2.5}$ pollution episodes in the winter of 2017. The ammonium to sulfate ratio has always been above 1.5 in Beijing, and has exceeded 3.0 since 2011. Sufficient ammonia provided strong atmospheric neutralization and weakened the particle acidity in Beijing, but the increased nitrate fraction was found to be causing the particle pH

elevation. During the campaign, the pH of $PM_{2.5}$ increased from 4.4 to 5.4 as the molar ratio of nitrate to sulfate increased from 1 to 5, which is firstly due to the less amount of sulfate, which suppressed the formation of H$^+$, and secondly due to the ammonia neutralization.

Sensitivity tests of particle hygroscopicity and acidity were conducted to investigate the possible changes in the

physiochemical properties if ammonia and nitrate are not well controlled in China in the future. The results showed

that the nitrate-rich particles can absorb more water than particles with higher sulfate fractions under a moderate

humid condition (RH<60%), and the particle pH increases rapidly due to the synergetic effect of ammonia and

nitrate, which will very likely occur in China in the upcoming years, because both of these pollutants are not well

controlled yet. The changes in particle pH and hygroscopicity will further enhance the uptake of gaseous compounds,

promote the chemical reactions which favor lower acidity, and also affect the optical properties of airborne particles

in China. Therefore, the processes and properties of haze particles during nitrated-dominated periods in the country

need to be thoroughly investigated with more consideration on the highly hygroscopic and neutralized particles.

**Data availability.**

Data can be accessed by contacting the corresponding author.

**Competing interests.**

The authors declare that they have no conflict of interest.

**Author contribution**

G.H. Wang conceived the study and designed the experiment. Y.N. Xie conducted the online IGAC-PM$_{2.5}$

chemical composition analysis and filter sampling in Beijing during the campaign. GQ. Tang, L.L. Wang, Y.S.

Wang and J. Gao provided other related observation data used in this article, including trace gases, PM mass

concentrations, meteorological data. G.H. Wang, X.P. Wang, Y.B. Chen, G.Y. Xue and S.S. Ge conducted the lab

analysis of filters and the data QA/QC. Y. N. Xie, G.H. Wang and J.M. Chen performed the data analysis. Y. N.

Xie and G.H. Wang wrote the paper. All the co-authors contributed to the data interpretation and discussion

**Acknowledgements**

This work was financially supported by the National Key R&D Program of China (2017YFC0210000), and the National Nature Science Foundation of China (No. 41773117). We thank Mr. Yicheng, Lin and Mr. Zhenrong Huang from Fortelice International Co., Ltd. for their technical supporting in IGAC operation during the campaign.

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

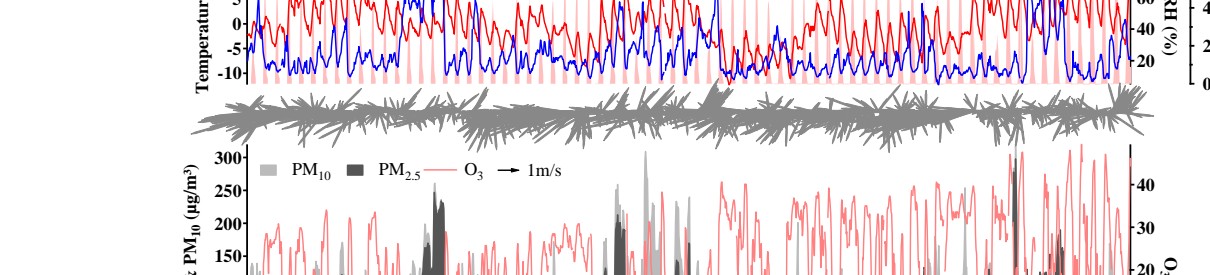

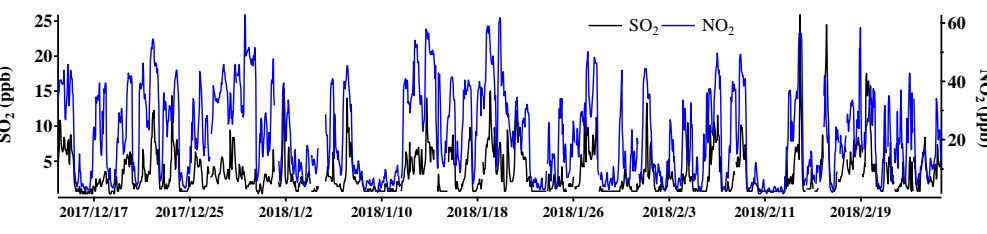

**FIG. 1. Timeseries of major pollutants during the campaign. Upper panel: radiation, temperature RH and wind arrows (drawn below); middle panel: PM₂.₅, PM₁₀ and ozone concentration; lower panel: SO₂ and NO₂ concentrations.**

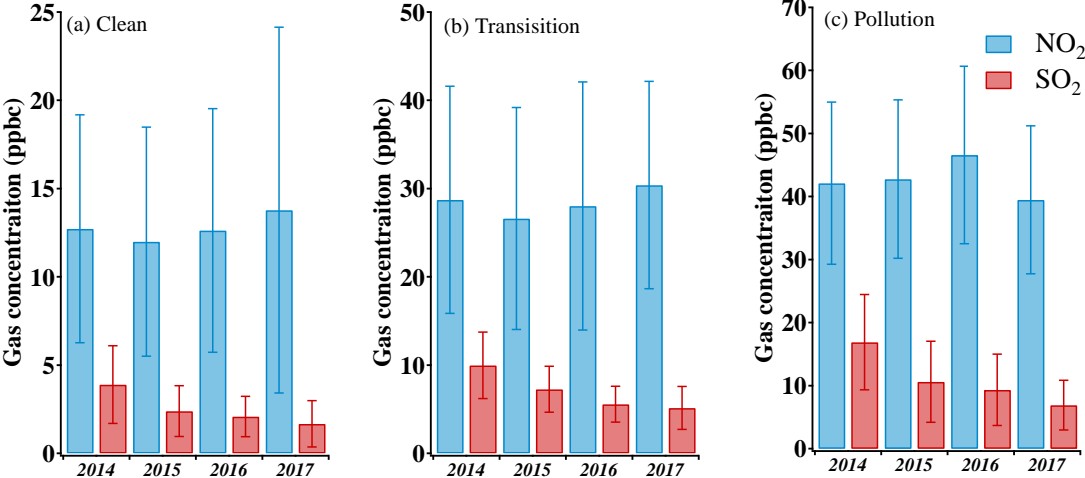


**FIG. 2. Statistics plot of NO₂ and SO₂ measured in downtown of Beijing at different PM₂.₅ levels during winter (December–February) for the past 4 years. Historical data (2014–2016) are from air quality real-time publishing platform, China National Environmental Monitoring Center, and data of 2017 is obtained during the campaign.**

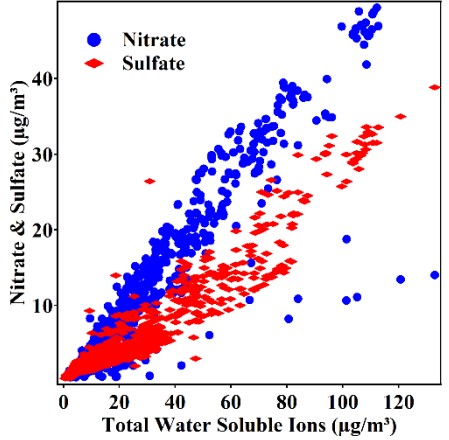

**FIG. 3. Scatter plots of nitrate and sulfate vs. WSIs during the campaign.**

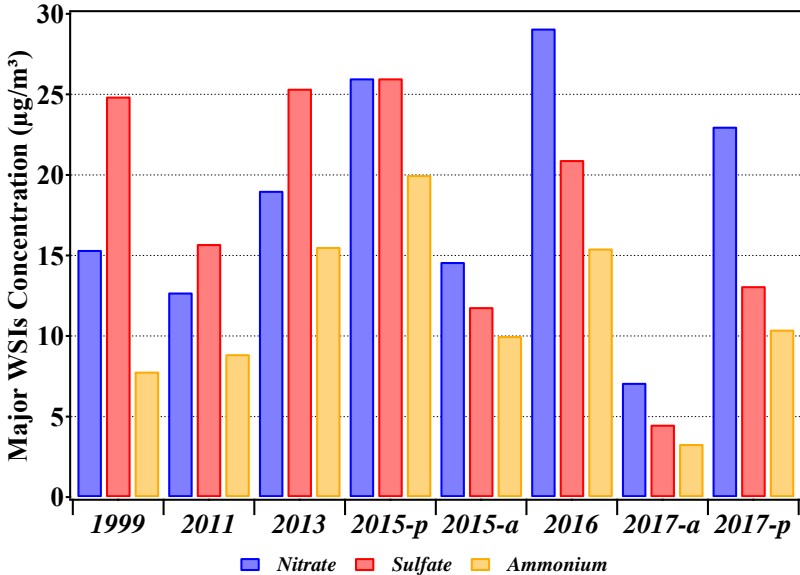

**FIG. 4. Major inorganic compositions in PM$_{2.5}$ observed in Beijing during winter from representative articles. The**

**results of 2017 denote the average concentration during haze episodes in this study. For details of the reviewed literature, please refer to Table S1.**

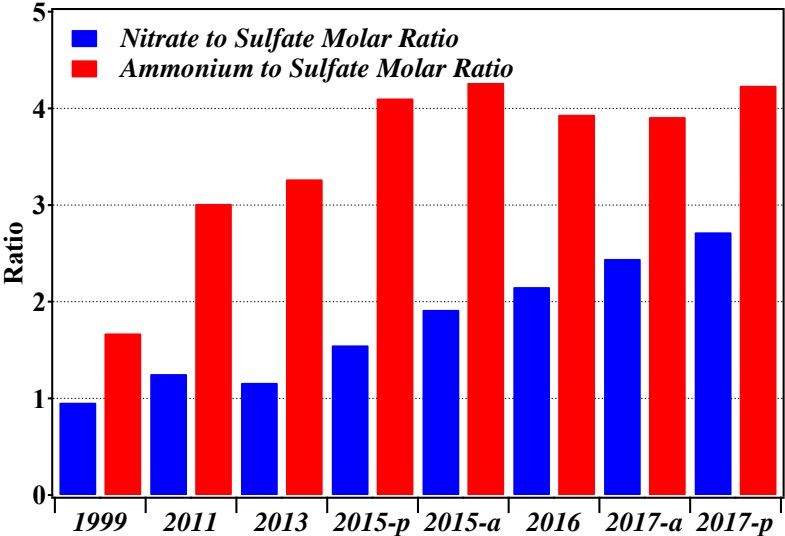

**FIG. 5. Ratio$_{N-to-S}$ and Ratio$_{A-to-S}$ calculated from the averaged data reported in representative research articles. Only the data in pollution episodes is chosen for 2017.**

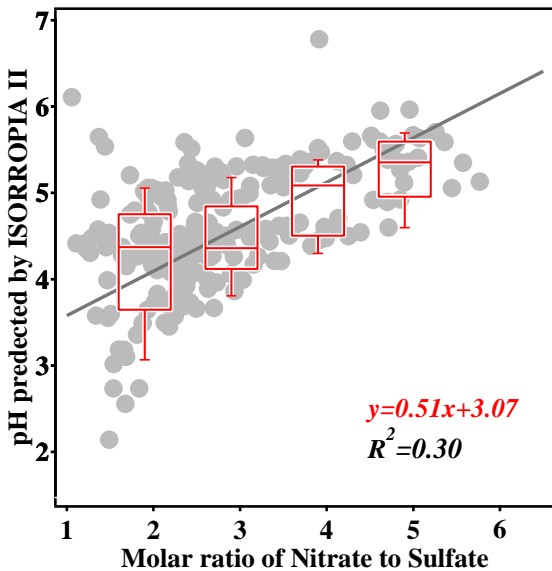

**FIG. 6. Scatter plot of simulated pH vs. Ratio_N-to-S. Linear fitting and the correlation coefficient are given. The box plot denotes the data points classified by 4 different ranges of the Ratio_N-to-S: 0–2;2–3;3–4; and 4–6. Only the data corresponding to the pollution categories and with sufficient aerosol liquid water (above 5 μg m⁻³) is chosen and the data during Chinese New Year is excluded.**

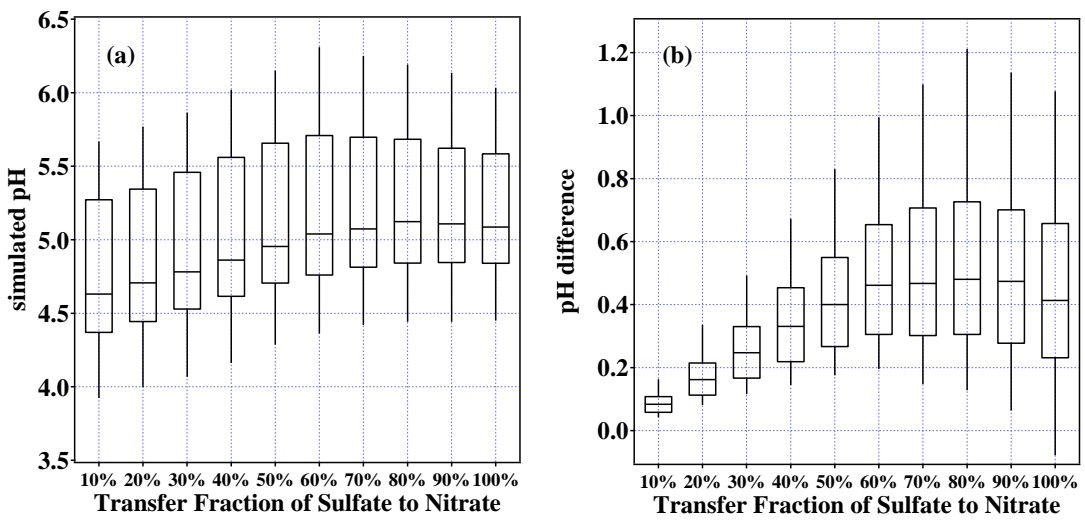

**Fig.7 Box plot of (a) simulated pH using the setting of the sensitivity test as well as (b) pH difference between the simulated pH from the transferred data and the original observation data (Only data with sufficient aerosol liquid water content (above 5 μg m⁻³) during the pollution period were shown).**

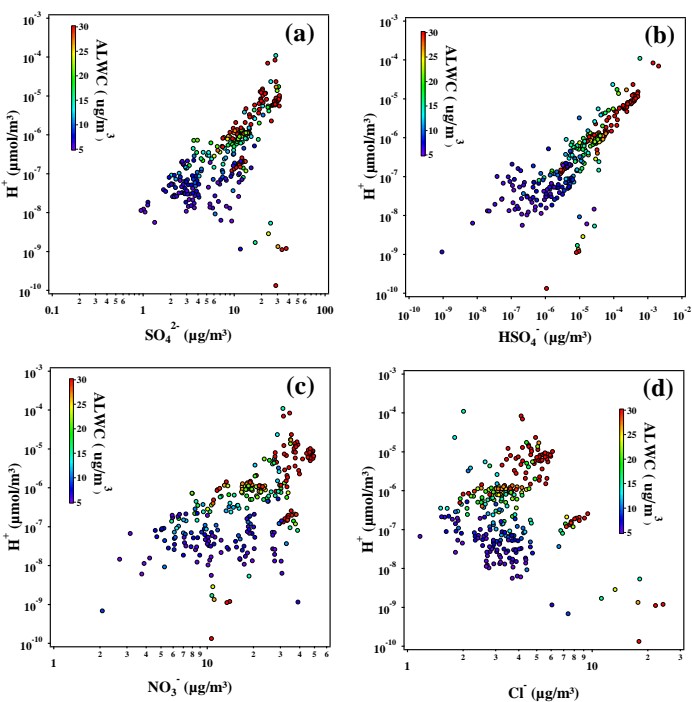

**FIG. 8. Scatter plots of simulated H⁺ ion vs. major inorganic anions, including (a) $SO_4^{2-}$, (b) $HSO_4^-$, (c)$NO_3^-$ and (d)$Cl^-$; the coordinates are in logarithm. Only the data corresponding to the pollution categories and with sufficient aerosol liquid water (above 5 μg m⁻³) was chosen and the data during Chinese New Year is excluded.**

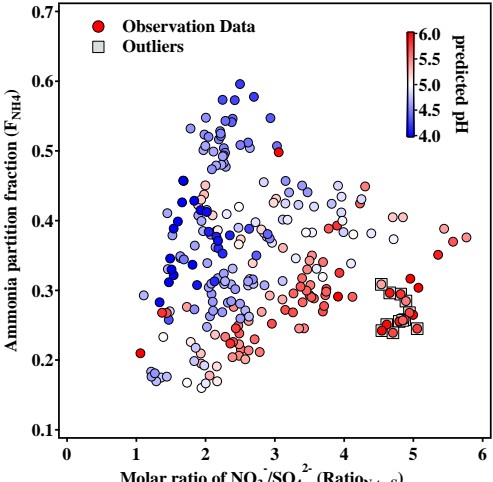

**FIG. 9. Scatter plot of $F_{NH4}$ vs. the $Ratio_{N-to-S}$ colored with predicted pH. Only the data corresponding to the pollution categories and with sufficient aerosol liquid water (above 5 μg m⁻³) is chosen and the data during Chinese New Year is excluded. Note that the grey frame depicts the outliers which have lower $F_{NH4}$ and lower Ammonia concentration.**

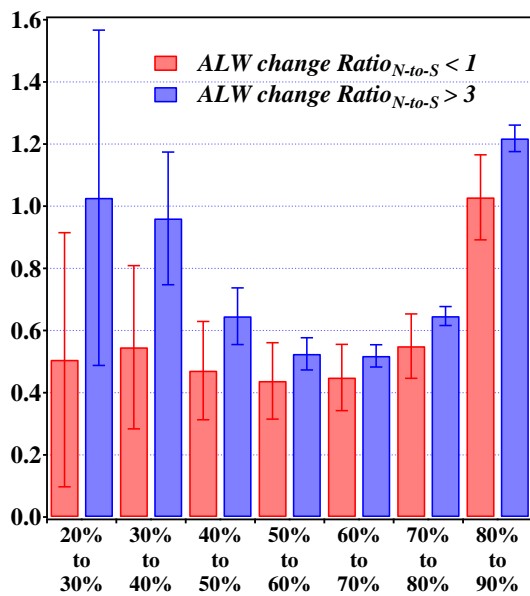

**FIG. 10. Relative ALWC change due to the RH elevation at different $Ratio_{N-to-S}$. Bars represent the relative change amount, and whiskers depict the standard deviation.**

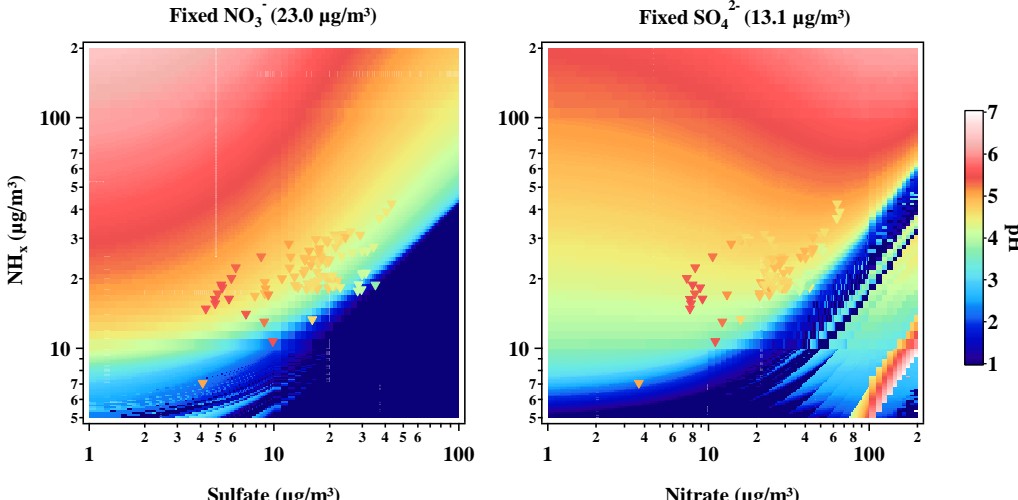

**FIG. 11**. **Sensitivity tests of pH's response to sulfate or nitrate change with inputting given NH$_x$, simulated by ISORROPIA II. Field measurement data (triangles) were drawn upon the simulation data and colored with predicted pH, respectively. The simulation is conducted with fixed RH (58%) and temperature (273.15 K).**

**Table 1.** Visibility and concentrations of major pollutants in Beijing during the winter of 2017

|  | Average | Clean | Transition | Pollution |
|---|---|---|---|---|
| Visibility (km) | 15±10 | 20±9.5 | 12±8.3 | 7.5±6.6 |
| PM$_{2.5}$ (µg/m$^3$) | 39.7±47.1 | 13.0±7.8 | 52.0±11.4 | 128±46.5 |
| PM$_{10}$ (µg/m$^3$) | 68.5±53.9 | 42.2±27.9 | 99.2±43.3 | 153.3±52.1 |
| O$_3$ (ppbv) | 18.5±12.8 | 23.6±11.2 | 10.6±10.0 | 8.4±10.3 |
| SO$_2$ (ppbv) | 3.2±3.1 | 1.7±1.3 | 5.1±2.5 | 6.9±1.3 |
| NO$_2$ (ppbv) | 21.4±14.8 | 13.8±10.4 | 32.8±9.6 | 39.8±11.6 |

**Table 2.** Concentrations(µg/m$^3$) of ammonia and inorganic ions of PM$_{2.5}$ measured by IGAC during the campaign

|  | Average | Clean | Transition | Pollution |
|---|---|---|---|---|
| NH$_3$ | 7.1±5.9 | 4.3±3.3 | 9.5±4.9 | 12.9±7.4 |
| Na$^+$ | 0.3±0.3 | 0.2±0.4 | 0.4±0.2 | 0.5±0.2 |

| | | | | |
|---|---|---|---|---|
| $NH_4^+$ | 3.3±4.4 | 0.9±0.8 | 3.7±2.4 | 10.4±4.8 |
| $K^+$ | 0.7±2.4 | 0.2±0.3 | 0.7±0.9 | 2.3±5.1 |
| $Mg^{2+}$ | 0.2±0.5 | 0.2±0.6 | 0.2±0.1 | 0.4±0.7 |
| $Ca^{2+}$ | 0.5±0.5 | 0.4±0.6 | 0.5±0.4 | 0.5±0.2 |
| $Cl^-$ | 2.4±2.3 | 0.9±0.8 | 2.3±0.8 | 4.6±2.9 |
| $NO_3^-$ | 7.1±9.6 | 1.7±1.4 | 7.9±3.2 | 23.0±10.7 |
| $SO_4^{2-}$ | 4.5±5.9 | 1.8±1.6 | 4.2±2.2 | 13.1±8.4 |