# Peer review of "Nitrate-dominated PM2.5 and elevation of particle pH observed in urban Beijing during the winter of 2017"

_Atmospheric Chemistry and Physics, 2019_

## Referee Comment (RC1) · Anonymous Referee #1 · 9 Sep 2019

General this paper investigates the pH of nitrate-dominated PM2.5. in Beijing in the winter of 2017. The acidity of particles is important in the discussion wether or not a S(IV) might be oxidized through NO2.

The English language of all the manuscript must be thoroughly checked and revised where needed. As the language correction alone is massive, I think the revision of the manuscript corresponds to 'major revision'.

Other than this , the manuscript is a solid work with interesting and valuable information and good analysis which should not be missed when Beijing wintertime sulphate formation is discussed.

Details

Title: Maybe better 'nitrate-dominated' instead of 'nitrate dominant' ?

Abstract, line 18: Better use 'Compared to historical records...' - see above comment, the English language of all of the manuscript has to be thoroughly checked, prefereble by a professional editor or a native speaker.

Introduction: Needs to be fully language-edited. I cannot do this in my review. Note especially singular/plural use is wrong very often

line 55ff: Mayve the role of non-classical $H_2O_2$ formation possibly contributing to $S(VI)$ formation shoul dbe mentioned here.

line 183ff: How much of the observed pattern is due to weather conditions ? Is there a possibility to 'de-weather' these observations ?

Figures 5 & 6: Alltogether, this is the most intersting finding of the MS. As the nitrae/sulphate ration increases, pH is expected to increase

line 242: Give the correlation coefficient of the straight line plotted in Figure 6

line 308: Please do not start a paragraph like this.

Figure 2: Maybe use identical y-axis scaling for (a), (b) and (c) ?

---

## Referee Comment (RC2) · Anonymous Referee #2 · 16 Oct 2019

This article evaluates the ongoing changes in the pH and nitrate content of PM2.5 in Beijing as strict controls on sulfur sources are reducing particle sulfate. This is a timely and important topic and the article is within the scope of ACP, however the particle requires extensive revisions to help clarify the authors points and to make it a useful contribution to the literature.

The authors use a detailed set of pollution measurements made in Beijing from Dec 2017 – Feb 2018 and compare they typical values measured for various pollutants (e.g. NO2, SO2) to those measured in previous years in Beijing. The most important dataset for the calculation of the aerosol pH is the water-soluble gas and PM2.5 constituents by

IGAC, however the timeseries for these measurements are not shown in the manuscript or in the SI. Given that the campaign period was ten weeks (15/12/2017 – 25/02/2018), there should be ∼1700 hourly data points for each compound. However there only appears to be, at most, a few hundred data points in Figures 3, 6, 7, 8, 9. This implies that perhaps the data coverage of the IGAC measurements was not very extensive during the campaign period, or that the data quality were often not sufficient. It would be useful for the authors to provide more explanation about the amount of valid data used in their pH calculations and to what extent it can be viewed as representative of the entire winter season. In fact, in Figures 4 and 5, it seems that the hi-volume sampler data is used for nitrate and sulfate, rather than the hourly data – is this true and why?

One confusing aspect of the manuscript is that the authors consistently refer to aerosol with a pH of 5.4 as 'near neutral', despite the proton activity being ∼ 40 times higher than a solution with a truly neutral pH (i.e. 7). From this, I believe they mean that a pH value of 5.4 is close to what one calculates for a solution exposed to 400 ppm of CO2 in the ambient atmosphere. I suggest changing this language because 'neutral' has a very specific meaning, different from what is being used here. If the authors want to emphasize that the pH is close to what might be expected in the absence of high particle pollution, they could explain that a value of 5.5 is expected in 'unpolluted' conditions. However even at very low PM2.5 mass loadings, the contributions of solutes other than carbonic acid/bicarbonate will dominate the ion balance and set the pH and I do not think there is anything special about a pH of 5.4.

Section 3.4 addresses the main question of the publication – how changes in particle composition are linked to changes in particle pH. Because nitrate is a semi-volatile component of the particle, its gas-particle partitioning is sensitive to the particle pH (and to its LWC and temperature). Thus it does not necessarily make sense to frame the question as 'the effect of nitrate fraction elevation on particle acidity'. I would view it from the opposite perspective – for a given amount of total ammonia, less PM2.5

sulfate allows the particle pH to be higher, allowing for nitrate to be present in the particle phase. In other words, the pH is not responding to the nitrate to sulfate ratio, as is suggested by the choice of axes in Figure 6. Rather, the pH is responding to the reductions in sulfate and thus leading to a change in the partitioning of nitrate. This is the converse of the explanation provided on Lines 290-291.

In particular, the statement on Lines 253-254 is very confusing: 'Less predicted H+ ion in aerosol liquid water is found to be the major cause of the higher pH. . .' A lower concentration (or activity) of H+ is the definition of higher pH (pH = -log[H+]), not just a major cause! Similarly, there is no reason to examine the [HSO4-]/[SO42-] ratio to consider the 'aerosol's ability of excess H+ formation' (Line 270). This ratio derived from the model output is going to be self-consistent with the pH calculated by the model given that the [HSO4-]/[SO42-] ratio depends on the pKa of bisulfate and the pH of the aerosol liquid water. There isn't any additional insight provided by this ratio is you already know the pH, which is well above the pKa in almost all cases.

Generally, I found the frequent references to the mechanisms of sulfur oxidation scattered through the text to be distracting. It would be preferable to state in one section of the introduction how and why the pH might impact the sulfur oxidation mechanism and rate and then return to it in the discussion. Other mentions of it in the results section, e.g. Lines 244-247 are distracting because the observational data themselves do not evaluate this mechanism.

Section 4 – The authors assess the changes in hygroscopicity in more nitrate-rich particles by comparing the ALWC when the RH is increased by 10%. The authors should clarify whether these calculations were performed using the particle components only as inputs, or the particle and gas (e.g. NH3 and HNO3) components as well. This is because increasing the RH would also increase the gas-to-particle partitioning of the gases, so the increase in ALWC results not just from the increased water activity in the particle, but also from dissolving more solutes into the aqueous phase for semi-volatile constituents like nitrate and ammonium.

Specific comments Page 3, L77 – The Song et al., 2018 reference identified a coding error in ISORROPIA that led to unreliable results for calculations done for closed, stable systems. Several of the references discussed by the authors in this section used this approach for their calculations of pH and therefore it would useful for the authors to identify which of the papers may have reported pH values that are in need of revision.

Page 7, Lines 215-220 Why is the ratio of ammonium/sulfate of 1.5 set as the threshold or limit for nitrate formation? It should be possible to carry out a more sophisticated analysis of the threshold for nitrate formation than what was performed in Pathak et al. 2009 and 2011.

Page 7, Lines 230 -232 The Shah et al. and Weber et al. studies do not necessarily contradict each other as they each examine trends and sensitivities in pH in different seasons and regions of the U.S.

Page 8, Line 279 and Figure 9 - The authors use inconsistent language and definitions for the ratio of particle $NH4+$ to total ammonium (conversion ratio in Figure 9 and 'ammonia partition fraction' on Line 279.

Reference list - Wang 2016a and Wang 2016b are the same reference

---

## Author Comment (AC2) · 25 Dec 2019

We are grateful to the referee and made modification according to his/her comments. Listed below are the point-by-point replies to the comments. please see the revised version of the manuscript.

Best regards,

General comments: Comments: General this paper investigates the pH of nitrate-dominated PM2.5. in Beijing in the winter of 2017. The acidity of particles is important in the discussion whether or not a S(IV) might be oxidized through NO2. The English

language of all the manuscript must be thoroughly checked and revised where needed. As the language correction alone is massive, I think the revision of the manuscript corresponds to 'major revision'. Other than this, the manuscript is a solid work with interesting and valuable information and good analysis which should not be missed when Beijing wintertime sulphate formation is discussed. Response: We thank the referee for his/her kind reply. We have thoroughly checked and revised language. We hope that the current format of the manuscript is good enough in the aspect of language.

Detailed comments Comments: Title: Maybe better 'nitrate-dominated' instead of 'nitrate dominant'? Response: Suggestion taken. Please see the title on page 1.

Comments: Abstract, line 18: Better use 'Compared to historical records...' - see above comment, the English language of all of the manuscript has to be thoroughly checked, prefereble by a professional editor or a native speaker. Response: Suggestion taken. We have asked a professional editor to help us revise the language problem. We modified it with "Compare with . . .", please see page 1, line 18-19. Comments Introduction: Needs to be fully language-edited. I cannot do this in my review. Note especially singular/plural use is wrong very often Response: Suggestion taken, as mentioned above, we have asked a professional editor to help us revise the language problem. We have corrected most of the singular/plural misusing in the text For example, page 2, line 42.

Comments: line 55ff: Maybe the role of non-classical H2O2 formation possibly contributing to S(VI) formation should be mentioned here. Response: Yes, suggestion taken. Non-classical H2O2 formation pathways are important in various conditions, especially in the haze episodes of China. We have added some related introduction into the text. Please see page 3, line 50-61.

Comments: line 183ff: How much of the observed pattern is due to weather conditions? Is there a possibility to 'de-weather' these observations? Response: The referee gave us a very good future direction. The pollution -weather feedback might be more complicated than the chemistry itself only. However, we believe this topic is beyond the scope

of this paper. We would have further investigation on the 'de-weather' pattern's link to physiochemical properties of particles in next studies.

Comments: Figures 5 & 6: All together, this is the most interesting finding of the MS. As the nitrate/sulphate ratio increases, pH is expected to increase. Response: Thanks for the comment, and we have modified the discussion on the cause of it, please refer to page 12-13, line 270-288ãĂĆ

Comments: line 242: Give the correlation coefficient of the straight line plotted in Figure 6 line 308: Please do not start a paragraph like this. Response: We apologize for the inconvenience by the poor language, and revised the language accordingly. The correlation coefficient was added on the figure.

Comments: Figure 2: Maybe use identical y-axis scaling for (a), (b) and (c) ? Response: It should be a good idea to use identical axis in data comparison. However, if we use identical y-axis, then the trend might not be significant since the magnitude is quite different among the three situations.

Please also note the supplement to this comment:
https://www.atmos-chem-phys-discuss.net/acp-2019-541/acp-2019-541-AC2-supplement.pdf
* * *

---

## Author Response (AR1)

**Dear Editor,**

Thank you for your kind work on handling our manuscript. We apologize for the poor English text in the last version of our manuscript, and revised the text with the help of professional editing service. What is more important is the two referee's comments, we have carefully read through the comments and either reply it or make modifications accordingly. Our responses to the two referees' comments are itemized below. Please refer to the modifications in the revised manuscript. We also wonder if we could change the tittle of the paper to "Nitrate-dominated PM2.5 and elevation of particle pH observed in urban Beijing during the winter of 2017", since the original tittle might cause misunderstandings.

Anything about our manuscript, please feel free to contact us by ynxie@geo.ecnu.edu.cn (Dr.Yuning Xie) or ghwang@geo.ecnu.edu.cn (Prof. Gehui Wang)

Best regards!

Yuning Xie

12/25, 2019

**Response to Anonymous Referee #1**

**General comments:**

**Comments**: General this paper investigates the pH of nitrate-dominated  $PM_{2.5}$ . in Beijing in the winter of 2017. The acidity of particles is important in the discussion whether or not a S(IV) might be oxidized through NO2. The English language of all the manuscript must be thoroughly checked and revised where needed. As the language correction alone is massive, I think the revision of the manuscript corresponds to 'major revision'. Other than this, the manuscript is a solid work with interesting and valuable information and good analysis which should not be missed when Beijing wintertime sulphate formation is discussed.

**Response**: We thank the referee for his/her kind reply. We have thoroughly checked and revised language. We hope that the current format of the manuscript is good enough in the aspect of language.

**Detailed comments**

Comments: Title: Maybe better 'nitrate-dominated' instead of 'nitrate dominant'? Response: Suggestion taken. Please see the title on page 1.

**Comments**: Abstract, line 18: Better use 'Compared to historical records...' - see above comment, the English language of all of the manuscript has to be thoroughly checked, prefereble by a professional editor or a native speaker.

**Response:** Suggestion taken. We have asked a professional editor to help us revise the language problem. We modified it with "Compare with …", please see page 1, line 18-19.

**Comments** Introduction: Needs to be fully language-edited. I cannot do this in my review. Note especially singular/plural use is wrong very often

**Response**: Suggestion taken, as mentioned above, we have asked a professional editor to help us revise the language problem. We have corrected most of the singular/plural misusing in the text For example, page 2, line 42.

**Comments**: line 55ff: Maybe the role of non-classical  $H_2O_2$  formation possibly contributing to S(VI) formation should be mentioned here.

**Response**: Yes, suggestion taken. Non-classical  $H_2O_2$  formation pathways are important in various conditions, especially in the haze episodes of China. We have added some related

introduction into the text. Please see page 3, line 50-61.

**Comments**: line 183ff: How much of the observed pattern is due to weather conditions? Is there a possibility to 'de-weather' these observations?

**Response**: The referee gave us a very good future direction. The pollution -weather feedback might be more complicated than the chemistry itself only. However, we believe this topic is beyond the scope of this paper. We would have further investigation on the 'de-weather' pattern's link to physiochemical properties of particles in next studies.

**Comments**: Figures 5 & 6: All together, this is the most interesting finding of the MS. As the nitrate/sulphate ratio increases, pH is expected to increase.

**Response**: Thanks for the comment, and we have modified the discussion on the cause of it, please refer to page 12-13, line  $270-288_{\circ}$

**Comments**: line 242: Give the correlation coefficient of the straight line plotted in Figure 6 line 308: Please do not start a paragraph like this.

**Response**: We apologize for the inconvenience by the poor language, and revised the language accordingly. The correlation coefficient was added on the figure.

**Comments**: Figure 2: Maybe use identical y-axis scaling for (a), (b) and (c) ?

**Response**: It should be a good idea to use identical axis in data comparison. However, if we use identical y-axis, then the trend might not be significant since the magnitude is quite different among the three situations.

**Respongse to anonymous Referee #2**

**General comments:**

**Comments**: This article evaluates the ongoing changes in the pH and nitrate content of  $PM_{2.5}$  in Beijing as strict controls on sulfur sources are reducing particle sulfate. This is a timely and important topic and the article is within the scope of ACP, however the particle requires extensive revisions to help clarify the authors points and to make it a useful contribution to the literature.

**Response**: Suggestion taken. We have extensively revised the manuscript including language modification. We will present the details in the next point-to-point replies to your comments.

**Comments**: The authors use a detailed set of pollution measurements made in Beijing from Dec 2017 – Feb 2018 and compare their typical values measured for various pollutants (e.g.  $NO_2$ ,  $SO_2$ ) to those measured in previous years in Beijing. The most important dataset for the calculation of the aerosol pH is the water-soluble gas and PM2.5 constituents by IGAC, however the timeseries for these measurements are not shown in the manuscript or in the SI. Given that the campaign period was ten weeks (15/12/2017 - 25/02/2018), there should be ~1700 hourly data points for each compound. However there only appears to be, at most, a few hundred data points in Figures 3, 6, 7, 8, 9. This implies that perhaps the data coverage of the IGAC measurements was not very extensive during the campaign period, or that the data quality were often not sufficient. It would be useful for the authors to provide more explanation about the amount of valid data used in their pH calculations and to what extent it can be viewed

as representative of the entire winter season. In fact, in Figures 4 and 5, it seems that the hivolume sampler data is used for nitrate and sulfate, rather than the hourly data – is this true and why?

**Response:** We thank the reviewer for his/her concerning about the data points in our manuscript figures. Not all of the data points were shown in this paper. From section 3.3 it could be easily seen that the SNA composition of PM2.5 significantly changed during the pollution periods. Therefore, the main purpose of this article lies in the examination on the impact of nitrate fraction elevation on the pH of particles. During the winter of 2017, the pollution happened less than before and the weather was quite dry. Only data during pollutions were used to plot the referee mentioned figures. More to that, the data were chosen with a criterion of whether ALWC was sufficient. Therefore, it might seem much less data were shown in the paper but it was intended. The data measured with IGAC are hourly data. Several places in the text were modified, mainly the captions of figures. Please see page 30, line 640-642; page 31, line 645 -646; page 32, line 649-650.

**Comments: One confusing aspect of the manuscript is that the authors consistently refer to aerosol with a pH of 5.4 as 'near neutral', despite the proton activity being ~ 40 times higher than a solution with a truly neutral pH (i.e. 7). From this, I believe they mean that a pH value of 5.4 is close to what one calculates for a solution exposed to 400 ppm of CO2 in the ambient atmosphere. I suggest changing this language because 'neutral' has a very specific meaning, different from what is being used here. If the authors want to emphasize that the pH is close to what might be expected in the absence of high particle pollution, they could explain that a value of 5.5 is expected in 'unpolluted' conditions. However, even at very low PM2.5 mass loadings, the contributions of solutes other than carbonic acid/bicarbonate will dominate the ion balance and set the pH and I do not think there is anything special about a pH of 5.4.**

**Response:** Thanks for the suggestion. We revised the description to "less acidic" and "more neutralized". A pH of 5.4 is an especially important value in previous literature (Cheng et al. 2016; Wang et al.2016; Seinfeld et al.2006), which discussed the topic on whether NO2

promotes sulfate formation in China. Please see the related discussions in the paper.

**Comments**: Section 3.4 addresses the main question of the publication – how changes in particle composition are linked to changes in particle pH. Because nitrate is a semi-volatile component of the particle, its gas-particle partitioning is sensitive to the particle pH (and to its LWC and temperature). Thus, it does not necessarily make sense to frame the question as 'the effect of nitrate fraction elevation on particle acidity'. I would view it from the opposite perspective – for a given amount of total ammonia, less PM2.5 sulfate allows the particle pH to be higher, allowing for nitrate to be present in the particle phase. In other words, the pH is not responding to the nitrate to sulfate ratio, as is suggested by the choice of axes in Figure 6. Rather, the pH is responding to the reductions in sulfate and thus leading to a change in the partitioning of nitrate. This is the converse of the explanation provided on Lines 290-291. **Response:** We disagree with the reviewer on the above comments. It has been found that the decrease of sulfate would not inevitably lead to an increase in particle pH and nitrate. For example, combing the field measurements and thermodynamic model simulations, Webber et al have investigated the variation trends of chemical composition and pH of PM2.5 in the southeastern US during the past 15 years (Weber et al, 2016). They found that pH of PM2.5 in US has kept constant in the range of 0-2 and litter change in particle ammonium nitrate, although sulfate in the fine particles has significantly decreased from about 7  $\mu$ g/m3 in 1999 to  $2 \mu g/m^3$  in 2014. However, it is not the case in China. In the past five years many studies have found that along with the sharp decreased in sulfate concentrations due to SO2 emission controls, relative abundance of nitrate of PM2.5 in many cities of China has significantly increased (Ji et al., 2018; Wu et al., 2018), as those found in this study. Such a different variation trend of nitrate suggests that aerosol chemistry (e.g., acidity) in China especially in haze periods is different from that in US and other developed countries, which is the motivation of this work why we would like to investigate the impact of changes in chemical compositions of PM2.5 on particle acidity in Beijing. It's of our special interest to point the difference of how particle's acidity reacts to its chemical composition's change. This study highlights the difference of particle acidity calculated by the same method between China and

the U.S.

**Comments**: In particular, the statement on Lines 253-254 is very confusing: 'Less predicted H+ ion in aerosol liquid water is found to be the major cause of the higher pH...' A lower concentration (or activity) of H+ is the definition of higher pH (pH = -log[H+]), not just a major cause! Similarly, there is no reason to examine the [HSO4-]/[SO42-] ratio to consider the 'aerosol's ability of excess H+ formation' (Line 270). This ratio derived from the model output is going to be self-consistent with the pH calculated by the model given that the [HSO4-]/[SO42-] ratio depends on the pKa of bisulfate and the pH of the aerosol liquid water. There isn't any additional insight provided by this ratio is you already know the pH, which is well above the pKa in almost all cases.

**Response:** We have reconsidered the comment by referee #2 and take it carefully. Our first intention is to use the ratio as a proxy of H+ production and to find the physiochemical nature by comparing the proxy to nitrate/sulfate ratio. However, after studying the referee#2's comment, we found that the ratio calculated from the result of ISORROPIA II model is almost equivalent to the pH, and thus Fig.8 is equivalent to Fig.6. Therefore, this figure and related discussion were deleted from the text. Besides this, we present more discussion on the effect of nitrate fraction elevation on NH3 partition ratio and its potential effect in the following part. The analysis shows that the partition of NH3 is more sensitive to nitrate content, and it is caused by the enhanced nitric acid partitioning due to higher particle pH. The deleted context were in section 3.4, before the paragraph in page 12, line 270. The added discussion were mainly in the last part of section 3.4, please refer to page12-13, line 270-288.

**Comments**: Generally, I found the frequent references to the mechanisms of sulfur oxidation scattered through the text to be distracting. It would be preferable to state in one section of the introduction how and why the pH might impact the sulfur oxidation mechanism and rate and then return to it in the discussion. Other mentions of it in the results section, e.g. Lines 244-247 are distracting because the observational data themselves do not evaluate this

mechanism.

**Response**: Suggestion taken. We added some introduction about SO2 oxidation into the introduction section and shortened the NO2 oxidation mechanism discussion in the results. See page 3, line 50-61; page 11, line 246-248;

**Comments**: Section 4 – The authors assess the changes in hygroscopicity in more nitrate-rich particles by comparing the ALWC when the RH is increased by 10%. The authors should clarify whether these calculations were performed using the particle components only as inputs, or the particle and gas (e.g. NH3 and HNO3) components as well. This is because increasing the RH would also increase the gas-to-particle partitioning of the gases, so the increase in ALWC results not just from the increased water activity in the particle, but also from dissolving more solutes into the aqueous phase for semi-volatile constituents like nitrate and ammonium.

**Response**: Suggestion taken. In our calculation, input  $NH_3$  was set as  $NH_3$  plus  $NH_4^+$ . But the input of  $HNO_3$  was only nitrate measured, since there were no  $HNO_3(g)$  measurements. As a result, there will only be as much  $NH_4NO_3$  as measured and the increase of ALWC is not from more solutes.

**Specific comments:**

**Comments**: Page 3, L77 – The Song et al., 2018 reference identified a coding error in ISORROPIA that led to unreliable results for calculations done for closed, stable systems. Several of the references discussed by the authors in this section used this approach for their calculations of pH and therefore it would useful for the authors to identify which of the papers may have reported pH values that are in need of revision.

**Response**: Right now, we couldn't get the fixed code to run the model in a more proper way. We would have further investigation on the model coding error in future work.

Comments: Page 7, Lines 215-220, Why is the ratio of ammonium/sulfate of 1.5 set as the

threshold or limit for nitrate formation? It should be possible to carry out a more sophisticated analysis of the threshold for nitrate formation than what was performed in Pathak et al.2009 and 2011.

**Response**: The RatioA-to-S is set to 1.5 by the definition of "excess ammonium", meaning that there was enough ammonia to form ammonium nitrate. Many field observations on the Chinese atmospheric aerosols including the work reported by Pathak et al,. (2004) found that nitrate aerosols can be significantly detected only when molar ratio of ammonium to sulfate is larger than 1.5. Actually, there was a quite comprehensive analysis based on experiments, please refer to Pathak et al, 2004. Thus, we think it is not necessary to repeat the analysis on this threshold, which was already done by Pathak et al. We have cited this work and readers can refer to this paper for the details.

**Comments**: Page 7, Lines 230 -232 The Shah et al. and Weber et al. studies do not necessarily contradict each other as they each examine trends and sensitivities in pH in different seasons and regions of the U.S.

Response: We found it inappropriate and rephrased the sentence. See page11, line235-237

Comments: Page 8, Line 279 and Figure 9 - The authors use inconsistent language and definitions for the ratio of particle  $NH_4^+$  to total ammonium (conversion ratio in Figure 9 and 'ammonia partition fraction' on Line 279.

**Response**: We've checked through the article and made revisions to make the language consistent on fig.9 (now as fig.8) and on page 13, line279-288.

**Comments**: Reference list - Wang 2016a and Wang 2016b are the same reference

**Response**: Thanks for the referee, we've modified this part. Other repeated reference like Cheng et al. 2016, Guo et al. 2017. See Page 19, line 417-419; Page 20, line 454 – 455; Page 25, line 560-565.

References:

- Cheng, Y., Zheng, G., Wei, C., Mu, Q., Zheng, B., Wang, Z., Gao, M., Zhang, Q., He, K., Carmichael, G., Pöschl, U., and Su, H.: Reactive nitrogen chemistry in aerosol water as a source of sulfate during haze events in China, Sci. Adv., 2, e1601530, 10.1126/sciadv.1601530, 2016.
- Wang, G., Zhang, R., Gomez, M. E., Yang, L., Levy Zamora, M., Hu, M., Lin, Y., Peng, J.,
  Guo, S., Meng, J., Li, J., Cheng, C., Hu, T., Ren, Y., Wang, Y., Gao, J., Cao, J., An, Z.,
  Zhou, W., Li, G., Wang, J., Tian, P., Marrero-Ortiz, W., Secrest, J., Du, Z., Zheng, J.,
  Shang, D., Zeng, L., Shao, M., Wang, W., Huang, Y., Wang, Y., Zhu, Y., Li, Y., Hu, J.,
  Pan, B., Cai, L., Cheng, Y., Ji, Y., Zhang, F., Rosenfeld, D., Liss, P. S., Duce, R. A., Kolb,
  C. E., and Molina, M. J.: Persistent sulfate formation from London Fog to Chinese haze, P.
  NATL. ACAD. SCI. USA, 113, 13630, 10.1073/pnas.1616540113, 2016.
- Seinfeld, J. H. and S. N. Pandis. Atmospheric chemistry and physics: from air pollution to climate change, John Wiley & Sons. ISBN: 978-1-118-94740-1. 2016
- Ji, D., et al. Characterization and source identification of fine particulate matter in urban Beijing during the 2015 Spring Festival. Sci. Total Environ. 628, 430-440, 2018.
- Pathak, R. K., Yao, X., and Chan, C. K.: Sampling Artifacts of Acidity and Ionic Species in PM2.5, Environ. Sci. Technol., 38, 254-259, 10.1021/es0342244, 2004.
- Webber et al. High aerosol acidity despite declining atmospheric sulfate concentrations over the past 15 years. Nature Geoscience, 2016, Vol.9, 282-286, DOI: 10.1038/NGEO2665
- Wu, C., et al. Chemical characteristics of haze particles in Xi'an during Chinese Spring Festival: Impact of fireworks burning. Journal of Environmental Sciences, Vol. 71, 179-187, 2018.

| Observation of nitrate dominantNitrate-dominated PM2.5 and                    |
|-------------------------------------------------------------------------------|
| elevation of particle pH <del>elevation</del>observed in urban Beijing |
| during the winter of 2017–                                                    |

Yuning Xie1, Gehui Wang 1,2, \*, Xinpei Wang 1, Jianmin Chen2,3, Yubao Chen 1, Guiqian Tang 4, \*Lili Wang 4, Shuangshuang Ge 1, Guoyan Xue 1, Yuesi Wang4, Jian Gao5

5

1Key Laboratory of Geographic Information Science of the Ministry of Education, School of Geographic Sciences,4 East China Normal University, Shanghai 200241, China
2Institute of Eco-Chongming, 3663 N. Zhongshan Rd., Shanghai 200062, China

3Department of Environmental Science and Technology, Fudan University, Shanghai, China
 4State Key Laboratory of Atmospheric Boundary Layer Physics and Atmospheric Chemistry, Institute of Atmospheric Physics, Chinese Academy of Sciences, Beijing 100080, China
 5Chinese Research Academy of Environmental Sciences, Beijing 100000, China;
 *Correspondence to*: Prof. Gehui Wang (ghwang@geo.ecnu.edu.cn)

15 Abstract.-Particle acidity is crucial: Chinese government has exerted strict emission controls to understandsecondary formation processes in mitigate air pollution events since acidity has substantial impacts2013, which resulted in significant decreases in the concentrations of air pollutants such as SO2, NOx and PM2.5. To investigate the impact of such changes on the physiochemical properties of PM2.5. Quantification of particle acidity withsimulated pH yielded various values range from 0.7 and conflicting conclusions on sulfate formation-

| +            | Style Definition: Heading 2                                              |  |  |
|--------------|--------------------------------------------------------------------------|--|--|
| 1            | Style Definition: Heading 3                                              |  |  |
| Ì,           | Style Definition: Heading 4                                              |  |  |
|              | Style Definition: EndNote Category Heading: Check spelling and grammar   |  |  |
| Ľ,           | Style Definition: Authors                                                |  |  |
|              | Style Definition: EndNote Bibliography Title: Check spelling and grammar |  |  |
|              | Style Definition: EndNote Bibliography: Check spelling and grammar       |  |  |
| 11           | Style Definition: Affiliation                                            |  |  |
| .11
11    | Style Definition: MS title                                               |  |  |
|              | Style Definition: Revision                                               |  |  |
|              | Style Definition: Correspondence                                         |  |  |
|              | Style Definition: Title                                                  |  |  |
|              | Style Definition: Normal (Web)                                           |  |  |
|              | Style Definition: Subtitle: Left                                         |  |  |
|              | Style Definition: Header: Font: (Default) DengXian                       |  |  |
| 411
111   | Style Definition: Balloon Text: Font:                                    |  |  |
| WII
MIV   | Style Definition: Caption                                                |  |  |
| 1111
1117 | Style Definition: Comment Text: Font:                                    |  |  |
| 10
10     | Style Definition: Comment Subject: Font:                                 |  |  |
|              | Style Definition: Footer: Font: (Default) DengXian                       |  |  |
| 11           | Formatted                                                                |  |  |
|              | Formatted: Centered, Line spacing: 1.5 lines                             |  |  |
| 1            | Formatted: Space Before: 31.2 pt, Line spacing: 1.5 lines                |  |  |
| 1            | Formatted: Line spacing: 1.5 lines                                       |  |  |

| 20 | mechanismschemistry of atmospheric aerosols in China recently. In this article, we found that particle pH could                                         |
|----|---------------------------------------------------------------------------------------------------------------------------------------------------------|
|    | increase to near neutral (5.4) as a result of effective sulfur emission control. Benefit from strict conducted a                                        |
|    | comprehensive characterization on PM 2.5 in Beijing during the winter of 2017. Strict pollution control actions,                             |
|    | also reduced the average PM 2.5 concentration, reduced to a low level $\frac{\text{(of 39.7 µg/m^2)}}{\text{(m s)}}$ in urban Beijing during |
|    | the winter of 2017. Compare to historyCompared with the historical record (2014-2017), SO 2 -gradually                                       |
| 25 | decreased to a low level (of 3 2ppby2 ppby in 2017the winter) while of 2017 but NOs kept increasing level was                                           |

- 25 decreased to a low level (of 3.2ppbv2 ppbv in 2017the winter) while of 2017 but NO2 kept increasing level was still high (21.4ppbv4 ppbv in 2017the winter). As a response, nitrate's of 2017). Accordingly, contribution of nitrate (23.0 µg/m3 m-3) to PM2.5 become dominantfar exceeded over sulfate (13.1 µg/m3 m-3) during PM2.5the pollution episodes. The, resulting in a significant increase of nitrate to sulfate molar ratio significantly increased from 1 to 2.7 (value of 1999 and 2017). As nitrate's fraction significantly elevated, particle pH was also found to
- 30 increase in winter Beijing given sufficient ammonia (average concentration 7.1µg/m³, 12.9µg/m³ during pollution). During PM2.5 pollution episodes, the particle pH predicted. The thermodynamic model (ISORROPIA-II) calculation results showed that during the PM2.5 pollution episodes particle pH increased from 4.4 (moderate acidic) to 5.4 (near neutral) as nitrate to sulfatemore neutralized) when the molar ratio of nitrate to sulfate increased from 1 to 5-It is found that the major H+ contributor S(VI) was mostly in the form of sulfate, indicating
- 35 that anionsaerosols were more neutralized as the nitrate content elevated. In the final part, Based on the results of sensitivity tests, the future prediction offor the particle acidity change was discussed via sensitivity tests; On one hand, We found that nitrate\_rich particles would be being at low and moderate humid conditions (RH: 20%-50%) can absorb more twice amount of water compared to the then sulfate\_rich particles. This absorption contrast\_ doubles-, and the nitrate and ammonia with low to moderate RH (20%~50%). On the other hand, increased level of
- 40 nitrate and ammonia wouldhigher levels have synergetic effects leading to rapid elevation of, rapidly elevating particle pH to merely neutral (above 5.6). As moderate haze events might occur more frequently with highunder the abundant ammonia and particulate nitrate concentration\_dominated PM2.5 conditions, the major chemical

|    | processes during haze events and the control target shallshould be re-evaluated to obtain the most effective control                                       | (   | Formatted: Font color: Black                                          |
|----|------------------------------------------------------------------------------------------------------------------------------------------------------------|-----|-----------------------------------------------------------------------|
|    | strategy.                                                                                                                                                  | (   | Formatted: Font color: Black, English (United States)                 |
|    |                                                                                                                                                            |     |                                                                       |
| 45 | 1 Introduction •                                                                                                                                           | ~(  | Formatted: Font color: Black                                          |
|    |                                                                                                                                                            | ) [ | Formatted: Line spacing: 1.5 lines                                    |
|    | Severe haze had-pollution has been causing serious environmental problems and harming public health in                                                     | {   | Formatted: Left, Indent: First line: 0.74 cm, Line spacing: 1.5 lines |
|    | China over the past decades (He et al., 2001;Wang et al., 2016b;Zhang et al., 2015b). Strong(He et al. 2001; Wang                                          |     |                                                                       |
|    | et al. 2016; Zhang et al. 2015b). Therefore, strong actions werehave been taken to reverse improve the worsening                                           |     |                                                                       |
|    | atmospheric environment-situation, including cutting down the pollutants' emission emissions with forced                                                   |     |                                                                       |
| 50 | installation of catalytic converter on vehicles, buildings of building clean-coal power generation system, and-                                            |     |                                                                       |
|    | prohibition prohibiting open burning of crop residue burning induring the harvest seasons easons, etc. (Chen et al.,                                       |     |                                                                       |
|    | 2017;Zhang et al., 2012;Liu et al., 2016). (Chen et al. 2017; Zhang et al. 2012; Liu et al. 2016). As a result, the                                        |     |                                                                       |
|    | PM 2.5 pollutions werepollution is relieved to a level that metmeet the goals in AIR POLLUTION PREVENTION-                                      |     |                                                                       |
|    | AND CONTROL ACTION PLANAir Pollution Prevention and Control Action Plain (issued by the state-                                                             |     |                                                                       |
| 55 | councilState Council of China, http://www.gov.cn/zwgk/2013-09/12/content_2486773.htm, in Chinese). Among                                                   |     |                                                                       |
|    | all the regions of interests, Beijing has achieved great success in PM2.5 controlling (reduction (the annual average                                       |     |                                                                       |
|    | $PM_{2.5}$ concentration of 2017 was 58 $\mu$ g/m 3 m -3 ). Yet, compare to $PM_{2.5}$ concentration in Beijing is still higher than |     |                                                                       |

that in most developed countries, this record was still high. Despite emission control, it is also important toelucidate the key processes in atmosphere pollutions in Beijing and across China.

60 The cause of PM2.5 pollution in China was multivariate (Guo et al., 2014;Ding et al., 2013). One feature of PM2.5 pollution across the country is significant high secondary formation of inorganic components (Huang et al., 2014a). Sulfate, nitrate and ammonium (SNA) comprised over 30% of the PM25 mass, and SNA's fraction continues to increase during development of pollution episodes(Cao et al., 2012). While models could well predict the airborne particle pollutions in the U.S. or Europe, it is challenging to simulate the real atmospheric environment in China 65 (Wang et al., 2014;Ervens et al., 2003). Previous modeling works showed that the simulated PM2.s-concentrations were underestimated within current scheme, and suggested the importance of heterogeneous reactions in the SNA

formation processes (Huang et al., 2014b;Herrmann et al., 2005). In Beijing, severe haze events occur with abundant nitrogen species (NOx; NH3, etc.) and high RH while photochemistry is often less active (Wang et al., 2016b;Cheng et al., 2016b). Field observations, chamber experiments, source apportionments and simulation works all suggests

- that the joint effect of NO2, SO2, and NH3 is important in the sulfate formation processes in haze events (Cheng et al., 2016b;Wang et al., 2016b;He et al., 2018). Aqueous oxidation of SO2 by NO2 could be of major process of sulfate formation in winter Beijing, as well as transition metal ions (TMI) catalyzed oxidation (Wang et al., 2016b;Cheng et al., 2016b). Besides, though the photochemistry were less active during winter haze periods, extra radical provided by HONO might enhance the atmospheric oxidation capacity and thus leads to extra SNA formation (Tan et al., 2018). Since these reactions are all sensitive to particle acidity, adequate quantification of airborne particle's acidity
- 15 2010): Since these reactions are an sensitive to particle actually, adequate quantification of an other particle statisty is essential to elucidate the specific contribution.Particle acidity has widely been studied due to its important roles in haze formation, and is widely implemented in
- major models (Yu et al., 2005;Robert et al., 2016). Since there was rarely practical method to direct measure the acidity of particles in real atmosphere (Wei et al., 2018;Freedman et al., 2019), calculation of the particle pH by
   thermodynamic models had been the most used method to quantify particle acidity. Most models (ISORROPIA II, E-AIM-IV, AIOMFAC, etc.) can predict H+, ALWC, and partitioning of volatile/semi volatile components, such as
- ammonia (Fountoukis and Nenes, 2007;Clegg et al., 2008). These models' ability to describe physiochemical properties of airborne particles was validated by various studies (Weber et al., 2016;Guo et al., 2016;Shi et al., 2017;Tao and Murphy, 2019;Murphy et al., 2017). However, several publications using the same method gave
   different particle pH values in Beijing, and contradict conclusions were drawn on whether the sulfate formation by NO2 oxidation could be important. Cheng et al. conducted modeling work and suggest that Beijing's PM2.5-pH
- ranged from 5.4–6.2, which is favorable for aqueous NO2 oxidation. Not only by modeling works, field observation and chamber study also support the NO2 oxidation's major contribution and address the importance of high ALWC as well as sufficient ammonia (Wang et al., 2016a;Chen et al., 2019). Meanwhile, particle pH during winter of 2015
   to 2016 simulated by Liu et al with the same method was lower (3.0–4.9, average 4.2) and it was suggested that the
- acidic particle did not favor the NO2-oxidation mechanism. With averaged data from typical locations, Guo et al further concluded that high ammonia could not raise the particle pH high enough for the NO2 oxidation. On the other side, Song et al suggested that the model (ISORROPIA II) might have coding error that predict pH values to be negative or above 7. However, with lab studies and field observations, Wang et al raised concern that whether it was
   appropriate to elucidate the chemical reactions by particle pH predicted with only inorganic compositions. In fact, since the real atmosphere was affected by uncountable factors, it is common that particle pH would have variation
- when simulated with ambient data. At least moderate acidic to near neutral acidity was reported in China, and airborne particles were more neutralized than those in the US., given the fact that the gaseous ammonia was still at a high level to particulate ammonium (Song et al., 2018).-

[revised manuscript text omitted]

The observation was conducted at an urban site-the State Key Laboratory of Atmospheric Boundary Layer Physics and Atmospheric Chemistry, Institute of Atmospheric Physics (IAP), Chinese Academy of Sciences (39°58′28″N, 116°22′16″E) in Beijing. All the instruments were located on the roof of a two-story building. The main local emissions are mainly emitted from the vehicles, while and the industrial emission is much less greatly 165 reduced since the major factory/power plants wereare moved out of Beijing or phased out due to the emission control policy. Overall, this site represents a normal the typical atmospheric environment of urban Beijing, and from which the data obtained from this site is applicable to compare to can be compared with those from previous studies in Beijingthe city (Ji et al., 2018).

Field Code Changed

A continuous online measurement of atmospheric components was conducted with a time resolution of 1-hour.
Two 1405-TEOMTM continuous ambient connected to eitherPM monitors using PM2.5 or PM10 cyclone inlet (Metone)
was engagedwere applied to obtain PM2.5 and PM10 mass concentration.concentrations. For trace gases (O3, NO27
and SO2), a series of gas monitors were used for the hourly measurement (Model 49i, 42i and 43i, respectively).
Meteorology data, including ambient temperature, relative humidityRH, wind speed, wind direction and total solar
radiation, were measured with an automatic weather station (MILOS 520, VAISALA Inc., Finland) located in the
middle of the yard of the observation site yard. Visibility data fromof Beijing airport online data was obtainedwere
downloaded from the website of https://gis.ncdc.noaa.gov/maps/ncei/cdo/hourly. BesidesApart from these online
monitors, a high\_volume sampler (TISCH ENVIRONMENTAL) with a PM2.5 inlet was used to collect PM2.5 samples on a day/night basis (daytime of 8:00—17:50 and nighttime of 18:00—7:50).

The inorganic water-soluble components of PM2.5 (SO42-, NO3-, Cl-, NH4+, Na+, Ca2+, Mg2++ and K+) and 180 ammonia gas were measured with an online-IC system: IGAC (In-situ Gases and Aerosol Composition monitor, Fortelice International Co., Ltd.). IGAC is comprised comprises of two major2 parts: sampling unit and analysisunit(Young et al., 2016).analyzer unit (Young et al. 2016). A vertical wet annular denuder (WAD) was engagedis used to collect the gas-phase species prior to a scrub and impactor aerosol collector (SIC), while the latter part can efficiently collect particleparticles into liquid samples. During the campaign, 1mM H2O2 solution wasis used as the 185 absorption liquid for the air samples. Under most atmospheric conditions, the absorption liquid couldcan well absorb the  $\frac{targetedtarget}{target}$  atmospheric components (e.g. SO2). An ICS-5000+ ion chromatograph  $\frac{was engagedis}{target}$ used as the analyzer unit in this study. For anions, an AS18 column (2mm×250mm2 mm × 250 mm, Dionex™, Dionex™) IonPac™) wasis used while a CS-16 column (4mm×250mm4 mm × 250 mm, Dionex™ IonPac™) wasis chosen for the analysis ofto analyze major cations, both running with recommended eluent (solution of KOH for anion/-190 methane sulfonic acid for cation). The behaviorperformance of the system washas been tested and improved over recent years, and studies of  $PM_{2.5}$  water-soluble ionsion observations were have been conducted by using this-

8

instrumentit (Young et al., 2016;Song et al., 2018;Liu et al., 2017a).(Young et al. 2016; Song et al. 2018; Liu et al. 2017). A better sensitivity due to the advanced suppression technology of the system greatly enhancedenhances its ability to measure trace ions, such as sodium and magnesium, which eould beis important in studies of particle ion balance-studies. For details of the comparison between IGAC comparisons withand filter sampling results, please refer to supplementary materials (Fig.,S1).

**3. Results**

195

**3.1 Major pollutants' levels**

210 reported by Zhang et al. (2015a)). periods.

With strict control actions, there were less  $PM_{2.5}$  pollution occurred lessepisodes and theits concentration kept – – – Formatted: Line spacing: 1.5 lines at a low level during the most of the time in the winter of 2017. The average concentrations of  $PM_{2.5}$  and  $PM_{10}$  were

9

|     | $39.7_{\mu}g^{\mu}m^{3}m^{-3}$ and $68.5_{\mu}g^{\mu}m^{3}m^{-3}$ , respectivelyAccording to the PM 2.5 concentration, three conditions of the                   |
|-----|-----------------------------------------------------------------------------------------------------------------------------------------------------------------------------|
|     | atmospheric environment were classified in this study : clean (the PM2.5 was elassified: Clean below 35µg/m3;                                                               |
| 215 | Transition 35µg/m2 75µg/m2; Pollution above 75µg/m2. From clean, µg m-3), transition to polluted condition (the          |
|     | PM2.5 was between about 35 µg m-3 and 75 µg m-3) and pollution (the PM2.5 was above 75 µg m-3). In the clean, |
|     | transition and pollution periods , the average $PM_{2.5}$ concentration was concentrations were 13.0±7.8 µg/m 3 m -3 ,                         |
|     | $52.0\pm11.4 \ \mu g/m^2 \ m^{-3}$ and $128.0\pm46.5 \ \mu g/m^2 \ \mu g \ m^{-3}$ , respectively (as showed in Table 1), indicating that there was                         |
|     | still PM 2.5 pollutionspollution during this the winter $(, with a maximum hourly PM_{2.5} concentration was 298.0 \mu g/m2).$                                   |
| 220 | Average concentration of $298 \mu g  m^{-3}$ . The average ozone concentration was 18.5±12.8 ppbv, and the value decreased                                                  |
|     | as $PM_{2.5}$ concentrations increased. Average The average SO 2 concentration (3.2±3.1ppbv1 ppbv) was almost 10 times                                           |
|     | lower than that of NO 2 (21.4±14.8 <del>ppby). This</del> 8 ppby). The significant <del>contrast of</del> gap between SO 2 and NO 2        |
|     | concentration could concentrations can be attributed to the sulfur emission control over recent years and the fast                                                          |
|     | increase of gasoline vehicles in Beijing(Cheng et al. 2018; Wang et al. 2018b) (Cheng et al. 2018; Wang et al.                                                              |
| 225 | 2018b) Both of the two gaseous pollutants showed an increasing trend as the PMss concentration                                                                              |
| 225 | increases increased during the bare enjoyed shut the elevation increases of NO. level use much more significant thus                                                        |
|     | meterses increased during the naze episodes, but the elevation_increase of NO 2 iever was intern nore significant, thus                     |
|     | making it a more important role in the pontition processes.                                                                                                                 |
|     | $NO_2$ and $SO_2$ are the most important precursor gases for major secondary inorganic nitrate and sulfate in                                                               |
|     | PM 2.5 . Due to the emission control being effectively conducted, , the sulfur emission decreased significantly and                                              |
| 230 | thus led to , resulting in lower ambient SO 2 concentration. To better describe thisit, changes inof these the two                                        |
|     | precursor gases during winter wereare investigated by examining the data from 2014 -to 2016 in Beijing. Average                                                             |
|     | values and the standard deviation wereare plotted in Fig. 2. SO 2 showed a significant decreasing trend in all the                                               |
|     | three conditionsperiods. In 2014, the concentrations of SO 2 in the three periods were 3.9, 10.0 and 16.9 ppbv in the                                            |
|     | three conditions., respectively. The SO 2 concentration differences between each stage were getting-                                                             |

235 smaller.difference in different conditions was narrowing. Until 2017, the difference of SO2 concentrations in-

10

between any two of three conditions had been all three stages were within 10ppby10 ppby. Meanwhile, NO2 concentrations kept increasing after 2015 in clean and transition conditions, but in the NO2 concentration during the pollution periods of 2017 NO2-was surprisingly lower than that in 2014. Besides the fact thatAlthough the dilution condition was much better than before, more quick and strong actions were taken to prevent the PM2.5
 pollution in 2017, such as construction prohibition, private vehicle restriction and vast shutting down of factories in neighborhoodneighboring regions (Cheng et al., 2018). The significant drop of NO2 proves the effectiveness of pollution control actions(Cheng et al. 2018). The significant drop of NO2 proves the effectiveness of pollution in 2017.

**3.2 PM2.5 chemical compositions**

245 According to several previous reports, the chemical compositions of PM2.5 during the winter of 2017 in 5 Similar Beijing changed significantly (Shao et al., 2018; Elser et al., 2016; Ge et al., 2017; Huang et al., 2017; Wang et al., 2017).(Shao et al. 2018; Elser et al. 2016; Ge et al. 2017; Huang et al. 2017; Wang et al. 2017). The major inorganic ions of PM2.5 in Beijing during 2017the winter wereof 2017 included ammonium (3.3±4.4\_µg/m3 m-3), nitrate  $(7.1\pm9.6 \mu g/m^3 m^{-3})$ , sulfate  $(4.5\pm5.9 \mu g/m^3 m^{-3})$  and chloride  $(2.4\pm2.3 \mu g/m^3)$ . Increase  $m^{-3}$ ). 250 Concentrations of all the major compositions' concentration was observed components increased as the PM2.5 concentration increased, but changes in the crustal ion  $(Na^+, Mg^{2+}, and Ca^{2+})$  concentrations were less <del>differed in</del>different conditions. Potassiumsignificant among the clean, transition and pollution periods. K+ increased during the PM2.5 pollutions pollution episodes (average concentration:  $2.3\pm5.1 \,\mu\text{g/m}^2$  m-3), indicating the possible contribution from of biomass burning sources or fireworks during Chinese New Year. ChlorideCl: in PM2.5 has 255 been used as a tracer for biomass burning and the coal consumption-tracer. The concentration of Cl in PM2.5 chloride increased significantly as PM2.5 loading got higher, increased, but the imbalance of chloride molar concentration of chloride to potassium suggests that biomass burning might not be the major source of PM2.5

11

chloride in PM2.5 other than the coal consumption in Beijing'sduring the PM2.5 pollution\_episodes in Beijing.
\_\_\_\_\_SNA greatly increased the PM2.5 pollutionspollution. Unlike the findings from some previous studies (Wang et al., 2016b;Huang et al., 2014a; Ji et al., 2014a; Ji

**270 **3.3 Comparison of major inorganic compositions during the early 21st century in Beijing**

275

To illustrate the changes in chemical compositioncompositions of PM2.5 duringat the China's economy booming stage (1999–\_2017), nitrate, sulfate and ammonium wereare chosen for the comparison with previouspreviously reported data during winter in Beijing (Fig.\_4). Only winter-averaged winter-observation data or representative pollution records were chosenare selected to show the significant change of SIA compositioncompositions. On average, thoughalthough the concentration might be varied due to emissiondifferent emissions and weather condition variationsconditions over the years, SIA concentrations concentration in the winter of 2017 werewas the lowest compare tocompared with the years before. Sulfate concentration varied from 4.5\_µg/m3 m-3 to 25.4\_µg/m3 m-3 and contributed the most PM2.5 masses among SIA compositionsspecies during pollutionsthe pollution episodes before 2015. The emission control of SO2 started at the year ofin 2006 to prevent

| 280 | adverse atmospheric environment events such as acid rain and high particulate matter loading (Wang et al.,                                                                                                                                                                                                                                                                                                                                                                                                                                                                                                                                                                                                                                                                                                                                                                                                                                                                                                                                                                                                                                                                                                                                                                                                                                                                                                                                                                                                                                                                                                                                                                                                                                                                                                                                                                                                                                                                                                                                                                                                                                                                                                                                                                                                                                                                                                                                                                                                                                                                                                                                                                                                                                                                                                                                                                                                                                                                                                                                                                                                                                                  |
|-----|-------------------------------------------------------------------------------------------------------------------------------------------------------------------------------------------------------------------------------------------------------------------------------------------------------------------------------------------------------------------------------------------------------------------------------------------------------------------------------------------------------------------------------------------------------------------------------------------------------------------------------------------------------------------------------------------------------------------------------------------------------------------------------------------------------------------------------------------------------------------------------------------------------------------------------------------------------------------------------------------------------------------------------------------------------------------------------------------------------------------------------------------------------------------------------------------------------------------------------------------------------------------------------------------------------------------------------------------------------------------------------------------------------------------------------------------------------------------------------------------------------------------------------------------------------------------------------------------------------------------------------------------------------------------------------------------------------------------------------------------------------------------------------------------------------------------------------------------------------------------------------------------------------------------------------------------------------------------------------------------------------------------------------------------------------------------------------------------------------------------------------------------------------------------------------------------------------------------------------------------------------------------------------------------------------------------------------------------------------------------------------------------------------------------------------------------------------------------------------------------------------------------------------------------------------------------------------------------------------------------------------------------------------------------------------------------------------------------------------------------------------------------------------------------------------------------------------------------------------------------------------------------------------------------------------------------------------------------------------------------------------------------------------------------------------------------------------------------------------------------------------------------------------------|
|     | 2013;Wang et al., 2018b).(Wang et al. 2013; Wang et al. 2018b). As a response, result, the sulfate concentration in                                                                                                                                                                                                                                                                                                                                                                                                                                                                                                                                                                                                                                                                                                                                                                                                                                                                                                                                                                                                                                                                                                                                                                                                                                                                                                                                                                                                                                                                                                                                                                                                                                                                                                                                                                                                                                                                                                                                                                                                                                                                                                                                                                                                                                                                                                                                                                                                                                                                                                                                                                                                                                                                                                                                                                                                                                                                                                                                                                                                                                         |
| 285 | winter were decreasing-decreased gradually (see results of 1999, 2011, 2015-a and 2017), the average                                                                                                                                                                                                                                                                                                                                                                                                                                                                                                                                                                                                                                                                                                                                                                                                                                                                                                                                                                                                                                                                                                                                                                                                                                                                                                                                                                                                                                                                                                                                                                                                                                                                                                                                                                                                                                                                                                                                                                                                                                                                                                                                                                                                                                                                                                                                                                                                                                                                                                                                                                                                                                                                                                                                                                                                                                                                                                                                                                                                                                                        |
|     | concentration in recent years were has been much lower than that in the early 2000s -(detailed literature                                                                                                                                                                                                                                                                                                                                                                                                                                                                                                                                                                                                                                                                                                                                                                                                                                                                                                                                                                                                                                                                                                                                                                                                                                                                                                                                                                                                                                                                                                                                                                                                                                                                                                                                                                                                                                                                                                                                                                                                                                                                                                                                                                                                                                                                                                                                                                                                                                                                                                                                                                                                                                                                                                                                                                                                                                                                                                                                                                                                                                                   |
|     | comparison eouldcan be found at Lang et al. (2017)).in Lang et al. (2017)). However, it was widely reported that                                                                                                                                                                                                                                                                                                                                                                                                                                                                                                                                                                                                                                                                                                                                                                                                                                                                                                                                                                                                                                                                                                                                                                                                                                                                                                                                                                                                                                                                                                                                                                                                                                                                                                                                                                                                                                                                                                                                                                                                                                                                                                                                                                                                                                                                                                                                                                                                                                                                                                                                                                                                                                                                                                                                                                                                                                                                                                                                                                                                                                            |
|     | sulfate still contributed the most $\frac{1}{1000}$ PM 2.5 mass concentration during the severe haze $\frac{1}{1000}$ such as                                                                                                                                                                                                                                                                                                                                                                                                                                                                                                                                                                                                                                                                                                                                                                                                                                                                                                                                                                                                                                                                                                                                                                                                                                                                                                                                                                                                                                                                                                                                                                                                                                                                                                                                                                                                                                                                                                                                                                                                                                                                                                                                                                                                                                                                                                                                                                                                                                                                                                                                                                                                                                                                                                                                                                                                                                                                                                                                                                                                                    |
|     | in the winter of 2013 (Huang et al., 2014a;Guo et al., 2014;Ji et al., 2014). Heterogeneous(Huang et al. 2014a;                                                                                                                                                                                                                                                                                                                                                                                                                                                                                                                                                                                                                                                                                                                                                                                                                                                                                                                                                                                                                                                                                                                                                                                                                                                                                                                                                                                                                                                                                                                                                                                                                                                                                                                                                                                                                                                                                                                                                                                                                                                                                                                                                                                                                                                                                                                                                                                                                                                                                                                                                                                                                                                                                                                                                                                                                                                                                                                                                                                                                                             |
|     | Guo et al. 2014; Ji et al. 2014). The heterogeneous formation might be responsible for the enhanced                                                                                                                                                                                                                                                                                                                                                                                                                                                                                                                                                                                                                                                                                                                                                                                                                                                                                                                                                                                                                                                                                                                                                                                                                                                                                                                                                                                                                                                                                                                                                                                                                                                                                                                                                                                                                                                                                                                                                                                                                                                                                                                                                                                                                                                                                                                                                                                                                                                                                                                                                                                                                                                                                                                                                                                                                                                                                                                                                                                                                                                         |
|     | $\frac{\text{transfer}_{CONVERSION}}{\text{transfer}_{CONVERSION}} \text{ ratio } \frac{\text{of} \underline{from}}{\text{SO}_2} \text{ to particulate sulfate}_{\underline{s}} \text{ including the NO}_{2\underline{-}} promoted aqueous reaction and } \frac{1}{2} $ |
|     | transitionmetalcatalyzed oxidations (Huang et al., 2014b;Xie et al., 2015a).(Huang et al. 2014b; Xie et al                                                                                                                                                                                                                                                                                                                                                                                                                                                                                                                                                                                                                                                                                                                                                                                                                                                                                                                                                                                                                                                                                                                                                                                                                                                                                                                                                                                                                                                                                                                                                                                                                                                                                                                                                                                                                                                                                                                                                                                                                                                                                                                                                                                                                                                                                                                                                                                                                                                                                                                                                                                                                                                                                                                                                                                                                                                                                                                                                                                                                                                  |
| 290 | 2015). On the other hand, the NOx emission in northNorth China significantly increased as the amount of power                                                                                                                                                                                                                                                                                                                                                                                                                                                                                                                                                                                                                                                                                                                                                                                                                                                                                                                                                                                                                                                                                                                                                                                                                                                                                                                                                                                                                                                                                                                                                                                                                                                                                                                                                                                                                                                                                                                                                                                                                                                                                                                                                                                                                                                                                                                                                                                                                                                                                                                                                                                                                                                                                                                                                                                                                                                                                                                                                                                                               |
|     | consumption and transportation kept increasing. Therefore, nitrate in PM 2.5 had been increasing since 2011.                                                                                                                                                                                                                                                                                                                                                                                                                                                                                                                                                                                                                                                                                                                                                                                                                                                                                                                                                                                                                                                                                                                                                                                                                                                                                                                                                                                                                                                                                                                                                                                                                                                                                                                                                                                                                                                                                                                                                                                                                                                                                                                                                                                                                                                                                                                                                                                                                                                                                                                                                                                                                                                                                                                                                                                                                                                                                                                                                                                                                                     |
|     | Average The average concentration of nitrate rangedrose from $7.1 \mu g/m^3 m^{-3}$ to $29.1 \mu g/m^3 m^{-3}$ . By the year of                                                                                                                                                                                                                                                                                                                                                                                                                                                                                                                                                                                                                                                                                                                                                                                                                                                                                                                                                                                                                                                                                                                                                                                                                                                                                                                                                                                                                                                                                                                                                                                                                                                                                                                                                                                                                                                                                                                                                                                                                                                                                                                                                                                                                                                                                                                                                                                                                                                                                                                                                                                                                                                                                                                                                                                                                                                                                                                                                                                                                             |
|     | 2015, the nitrate concentration had exceeded the sulfate concentration, and the two equally contributed to PM                                                                                                                                                                                                                                                                                                                                                                                                                                                                                                                                                                                                                                                                                                                                                                                                                                                                                                                                                                                                                                                                                                                                                                                                                                                                                                                                                                                                                                                                                                                                                                                                                                                                                                                                                                                                                                                                                                                                                                                                                                                                                                                                                                                                                                                                                                                                                                                                                                                                                                                                                                                                                                                                                                                                                                                                                                                                                                                                                                                                                                               |
|     | mass equally as sulfate in pollutions in winter. pollution episodes. Although the nitrate's concentration during                                                                                                                                                                                                                                                                                                                                                                                                                                                                                                                                                                                                                                                                                                                                                                                                                                                                                                                                                                                                                                                                                                                                                                                                                                                                                                                                                                                                                                                                                                                                                                                                                                                                                                                                                                                                                                                                                                                                                                                                                                                                                                                                                                                                                                                                                                                                                                                                                                                                                                                                                                                                                                                                                                                                                                                                                                                                                                                                                                                                                                            |
| 295 | pollution periodperiods decreased in 2017 (23.0 $\mu$ g/m 2 m -3 ), the decrease was not obvious significant and the                                                                                                                                                                                                                                                                                                                                                                                                                                                                                                                                                                                                                                                                                                                                                                                                                                                                                                                                                                                                                                                                                                                                                                                                                                                                                                                                                                                                                                                                                                                                                                                                                                                                                                                                                                                                                                                                                                                                                                                                                                                                                                                                                                                                                                                                                                                                                                                                                                                                                                                                                                                                                                                                                                                                                                                                                                                                                                                                                                                                                  |
|     | concentration was still comparable to previous studies. As for The winter-averaged ammonium, the winter average                                                                                                                                                                                                                                                                                                                                                                                                                                                                                                                                                                                                                                                                                                                                                                                                                                                                                                                                                                                                                                                                                                                                                                                                                                                                                                                                                                                                                                                                                                                                                                                                                                                                                                                                                                                                                                                                                                                                                                                                                                                                                                                                                                                                                                                                                                                                                                                                                                                                                                                                                                                                                                                                                                                                                                                                                                                                                                                                                                                                                                             |
|     | concentration reached the maximum (~20_ $\mu$ g/m 3 m -3 ) in the year of 2015, but decreased afterwards. Taking the                                                                                                                                                                                                                                                                                                                                                                                                                                                                                                                                                                                                                                                                                                                                                                                                                                                                                                                                                                                                                                                                                                                                                                                                                                                                                                                                                                                                                                                                                                                                                                                                                                                                                                                                                                                                                                                                                                                                                                                                                                                                                                                                                                                                                                                                                                                                                                                                                                                                                                                                                                                                                                                                                                                                                                                                                                                                                                                                                                                                                  |
|     | role The decreasing trend of ammonium-as, which is the major neutralizer in the atmosphere of Beijing, the                                                                                                                                                                                                                                                                                                                                                                                                                                                                                                                                                                                                                                                                                                                                                                                                                                                                                                                                                                                                                                                                                                                                                                                                                                                                                                                                                                                                                                                                                                                                                                                                                                                                                                                                                                                                                                                                                                                                                                                                                                                                                                                                                                                                                                                                                                                                                                                                                                                                                                                                                                                                                                                                                                                                                                                                                                                                                                                                                                                                                                                  |
|     | decreasing trend-well represented the efficient pollution control of SNA compositioncompositions during the                                                                                                                                                                                                                                                                                                                                                                                                                                                                                                                                                                                                                                                                                                                                                                                                                                                                                                                                                                                                                                                                                                                                                                                                                                                                                                                                                                                                                                                                                                                                                                                                                                                                                                                                                                                                                                                                                                                                                                                                                                                                                                                                                                                                                                                                                                                                                                                                                                                                                                                                                                                                                                                                                                                                                                                                                                                                                                                                                                                                                                                 |
| 300 | winter of 2017. In a word, nitrate as the dominant composition-, high nitrate fraction has become one of the major                                                                                                                                                                                                                                                                                                                                                                                                                                                                                                                                                                                                                                                                                                                                                                                                                                                                                                                                                                                                                                                                                                                                                                                                                                                                                                                                                                                                                                                                                                                                                                                                                                                                                                                                                                                                                                                                                                                                                                                                                                                                                                                                                                                                                                                                                                                                                                                                                                                                                                                                                                                                                                                                                                                                                                                                                                                                                                                                                                                                                                          |
|     | features of PM 2.5 in Beijing during winter.                                                                                                                                                                                                                                                                                                                                                                                                                                                                                                                                                                                                                                                                                                                                                                                                                                                                                                                                                                                                                                                                                                                                                                                                                                                                                                                                                                                                                                                                                                                                                                                                                                                                                                                                                                                                                                                                                                                                                                                                                                                                                                                                                                                                                                                                                                                                                                                                                                                                                                                                                                                                                                                                                                                                                                                                                                                                                                                                                                                                                                                                                                     |

\_\_\_\_The ratio between major SIA compositions could components can better represent the composition change

|     | from data discussed above. As shown in figFig.5, the nitrate to sulfate molar ratio (Ratio N-to-S ) hadhas been                                                                                                                                                                                                                                                                                                                                                                                                                                                                                                                                                                                                                                                                                                                                                                                                                                                                                                                                                                                                                                                                                                                                                                                                                                                                                                                                                                                                                                                                                                                                                                                                                                                                                                                                                                                                                                                                                                                                                                                                     |                                       |
|-----|--------------------------------------------------------------------------------------------------------------------------------------------------------------------------------------------------------------------------------------------------------------------------------------------------------------------------------------------------------------------------------------------------------------------------------------------------------------------------------------------------------------------------------------------------------------------------------------------------------------------------------------------------------------------------------------------------------------------------------------------------------------------------------------------------------------------------------------------------------------------------------------------------------------------------------------------------------------------------------------------------------------------------------------------------------------------------------------------------------------------------------------------------------------------------------------------------------------------------------------------------------------------------------------------------------------------------------------------------------------------------------------------------------------------------------------------------------------------------------------------------------------------------------------------------------------------------------------------------------------------------------------------------------------------------------------------------------------------------------------------------------------------------------------------------------------------------------------------------------------------------------------------------------------------------------------------------------------------------------------------------------------------------------------------------------------------------------------------------------------------------------|---------------------------------------|
|     | increasing significantly from below 1.0 to 2.7 (1999 vs. 2017). Ratio N Ratio N-to-S was around 1 before 2013 but                                                                                                                                                                                                                                                                                                                                                                                                                                                                                                                                                                                                                                                                                                                                                                                                                                                                                                                                                                                                                                                                                                                                                                                                                                                                                                                                                                                                                                                                                                                                                                                                                                                                                                                                                                                                                                                                                                                                                                                        |                                       |
| 305 | then steadily increased after the year of 2013, same as previouslyprevious publications (Shao et al., 2018;Lang et-                                                                                                                                                                                                                                                                                                                                                                                                                                                                                                                                                                                                                                                                                                                                                                                                                                                                                                                                                                                                                                                                                                                                                                                                                                                                                                                                                                                                                                                                                                                                                                                                                                                                                                                                                                                                                                                                                                                                                                                                            |                                       |
|     | al., 2017). (Shao et al. 2018; Lang et al. 2017). Interestingly, Ratio N-to-S during pollution episodes was lower than                                                                                                                                                                                                                                                                                                                                                                                                                                                                                                                                                                                                                                                                                                                                                                                                                                                                                                                                                                                                                                                                                                                                                                                                                                                                                                                                                                                                                                                                                                                                                                                                                                                                                                                                                                                                                                                                                                                                                                                              | Formatted: Not Superscript/ Subscript |
|     | the winter average in 2015, but $\frac{\text{Ratio}_{N}\text{Ratio}_{N-40-S}}{\text{during pollution episodes greatly exceeded the average value-in}$                                                                                                                                                                                                                                                                                                                                                                                                                                                                                                                                                                                                                                                                                                                                                                                                                                                                                                                                                                                                                                                                                                                                                                                                                                                                                                                                                                                                                                                                                                                                                                                                                                                                                                                                                                                                                                                                                                                                                                          |                                       |
|     | 2017, showing that nitrate's the dominance inof nitrate in the present PM2.5 pollutions nowadays.pollution. The                                                                                                                                                                                                                                                                                                                                                                                                                                                                                                                                                                                                                                                                                                                                                                                                                                                                                                                                                                                                                                                                                                                                                                                                                                                                                                                                                                                                                                                                                                                                                                                                                                                                                                                                                                                                                                                                                                                                                                                                                |                                       |
|     | rapid increase of NRationary ratio from around 1 to nearly 3 was not only resulted from the result of sulfur                                                                                                                                                                                                                                                                                                                                                                                                                                                                                                                                                                                                                                                                                                                                                                                                                                                                                                                                                                                                                                                                                                                                                                                                                                                                                                                                                                                                                                                                                                                                                                                                                                                                                                                                                                                                                                                                                                                                                                                                                   | Formatted: Subscript           |
| 310 | emission control, itbut also indicates that from more nitrate was formed and enterpartitioning to the particle phase-                                                                                                                                                                                                                                                                                                                                                                                                                                                                                                                                                                                                                                                                                                                                                                                                                                                                                                                                                                                                                                                                                                                                                                                                                                                                                                                                                                                                                                                                                                                                                                                                                                                                                                                                                                                                                                                                                                                                                                                                          |                                       |
|     | via partitioning. Abundant ammonia in the Beijing's atmosphere wouldcan enhance the partitioning of nitric acid                                                                                                                                                                                                                                                                                                                                                                                                                                                                                                                                                                                                                                                                                                                                                                                                                                                                                                                                                                                                                                                                                                                                                                                                                                                                                                                                                                                                                                                                                                                                                                                                                                                                                                                                                                                                                                                                                                                                                                                                                |                                       |
|     | gas by forming ammonium nitrate. To identify whether the ammonia wasis sufficient, the ammonium to sulfate                                                                                                                                                                                                                                                                                                                                                                                                                                                                                                                                                                                                                                                                                                                                                                                                                                                                                                                                                                                                                                                                                                                                                                                                                                                                                                                                                                                                                                                                                                                                                                                                                                                                                                                                                                                                                                                                                                                                                                                                                     |                                       |
|     | ratio (Ratio Ratio Ratio Context Ratio Rat |                                       |
|     | experienced ammonia insufficient situation insufficiency during summer (Ammonium to sulfate ration Ratio A-to-S:                                                                                                                                                                                                                                                                                                                                                                                                                                                                                                                                                                                                                                                                                                                                                                                                                                                                                                                                                                                                                                                                                                                                                                                                                                                                                                                                                                                                                                                                                                                                                                                                                                                                                                                                                                                                                                                                                                                                                                                                               |                                       |
| 315 | less than 1.5), thus limitlimiting the formation and partitioning of nitrate into particle phase (Pathak et al.,                                                                                                                                                                                                                                                                                                                                                                                                                                                                                                                                                                                                                                                                                                                                                                                                                                                                                                                                                                                                                                                                                                                                                                                                                                                                                                                                                                                                                                                                                                                                                                                                                                                                                                                                                                                                                                                                                                                                                                                                               |                                       |
|     | 2009;Pathak et al., 2011). the particle phase (Pathak et al., 2004; Pathak et al. 2009; Pathak et al. 2011). However,                                                                                                                                                                                                                                                                                                                                                                                                                                                                                                                                                                                                                                                                                                                                                                                                                                                                                                                                                                                                                                                                                                                                                                                                                                                                                                                                                                                                                                                                                                                                                                                                                                                                                                                                                                                                                                                                                                                                                                                                          |                                       |
|     | Ratio A Ratio A-to-S in winter Beijing during winter was always above 1.5, the The lowest value appeared in 1999                                                                                                                                                                                                                                                                                                                                                                                                                                                                                                                                                                                                                                                                                                                                                                                                                                                                                                                                                                                                                                                                                                                                                                                                                                                                                                                                                                                                                                                                                                                                                                                                                                                                                                                                                                                                                                                                                                                                                                                         |                                       |
|     | (averaged Ratio A-to-S -averaged: 1.7), then the ratio reached higher levelincreased rapidly (above 3) after the year of                                                                                                                                                                                                                                                                                                                                                                                                                                                                                                                                                                                                                                                                                                                                                                                                                                                                                                                                                                                                                                                                                                                                                                                                                                                                                                                                                                                                                                                                                                                                                                                                                                                                                                                                                                                                                                                                                                                                                                                            |                                       |
|     | 2011 (red bars in Fig.5). In recent years, the ammonium Ratio Ago sulfate ratio s has reached around 4. This value is                                                                                                                                                                                                                                                                                                                                                                                                                                                                                                                                                                                                                                                                                                                                                                                                                                                                                                                                                                                                                                                                                                                                                                                                                                                                                                                                                                                                                                                                                                                                                                                                                                                                                                                                                                                                                                                                                                                                                                                                          | Formatted: Subscript           |
| 320 | typically observed in the eastern American America during winter, though the absolute concentration wasis much                                                                                                                                                                                                                                                                                                                                                                                                                                                                                                                                                                                                                                                                                                                                                                                                                                                                                                                                                                                                                                                                                                                                                                                                                                                                                                                                                                                                                                                                                                                                                                                                                                                                                                                                                                                                                                                                                                                                                                                                                 |                                       |
|     | higher in Beijing (Shah et al., 2018). (Shah et al. 2018). To sum up, these results suggest that the effective sulfur                                                                                                                                                                                                                                                                                                                                                                                                                                                                                                                                                                                                                                                                                                                                                                                                                                                                                                                                                                                                                                                                                                                                                                                                                                                                                                                                                                                                                                                                                                                                                                                                                                                                                                                                                                                                                                                                                                                                                                                                          |                                       |
|     | emission control and ammonia-rich atmosphere providedprovide the favorable environment for nitrate formation,                                                                                                                                                                                                                                                                                                                                                                                                                                                                                                                                                                                                                                                                                                                                                                                                                                                                                                                                                                                                                                                                                                                                                                                                                                                                                                                                                                                                                                                                                                                                                                                                                                                                                                                                                                                                                                                                                                                                                                                                                  |                                       |
|     | and eventually ehangingchange PM2.5 in Beijing from sulfate-dominated to nitrate-dominated type.                                                                                                                                                                                                                                                                                                                                                                                                                                                                                                                                                                                                                                                                                                                                                                                                                                                                                                                                                                                                                                                                                                                                                                                                                                                                                                                                                                                                                                                                                                                                                                                                                                                                                                                                                                                                                                                                                                                                                                                                                               |                                       |
| •   |                                                                                                                                                                                                                                                                                                                                                                                                                                                                                                                                                                                                                                                                                                                                                                                                                                                                                                                                                                                                                                                                                                                                                                                                                                                                                                                                                                                                                                                                                                                                                                                                                                                                                                                                                                                                                                                                                                                                                                                                                                                                                                                                |                                       |

| I   |                                                                                                                                                    |                                           |
|-----|----------------------------------------------------------------------------------------------------------------------------------------------------|-------------------------------------------|
|     | 3.4 Aerosol pH's response to PM2.5 nitrate fraction the elevation of nitrate fraction in PM2.5                                                     | Formatted: Line spacing: 1.5 lines        |
|     |                                                                                                                                                    |                                           |
| 325 | The shift from sulfate dominant to nitrate dominant PM 2.5 will further influence the secondary chemical processes                      |                                           |
|     | via changing physiochemical properties of aerosols, e.g. hygroscopicity and particle acidity. In a thorough study m                                |                                           |
|     | the U.S, despite the emission of NO x was well controlled, nitrate fraction in PM 2.5 didn't show a corresponding            |                                           |
|     | decreasing trend. This was caused by the elevated partitioning of nitric acid to particle phase in eastern America                                 |                                           |
| 220 | (Snan et al., 2018). Researchers implied that higher nitrate partition fraction could be attributed to increasing particle                         |                                           |
| 330 | pH, which is contradictory to some publication focusing on the particle pH s trend in the U.S (Weber et al., 2016).                                |                                           |
|     | The critical problem on particle pH was more controversial in China since its importance in elucidating the key                                    |                                           |
|     | atmospheric chemistry processes (Cheng et al., 2016b;Guo et al., 2016;Wang et al., 2016b;Weber et al., 2016;Guo                                    |                                           |
|     | et al., 2017a;Lu et al., 2017a). Therefore, it is necessary to study the response of particle acidity to the chemical                              |                                           |
|     | composition changes based on high resolution observation dataset.                                                                                  |                                           |
| 335 | The shift from sulfate-dominated to nitrate-dominated PM 2.5 further influences the secondary chemical                                  |                                           |
|     | processes via changing physiochemical properties of aerosols, e.g. hygroscopicity and particle acidity. In a thorough                              |                                           |
|     | study in the U.S, despite the well control of $NO_x$ emission, the nitrate fraction in $PM_{2.5}$ didn't show a corresponding                      |                                           |
|     | decreasing trend. It was caused by the elevated partitioning of nitric acid to the particle phase in the eastern America                           |                                           |
|     | (Shah et al. 2018). Researchers implied that higher nitrate partition fraction is resulted from the increasing particle                            |                                           |
| 340 | pH, while some studies showed that the particle pH was decreasing as particulate sulfate decreases in the U.S (Weber                               |                                           |
|     | et al. 2016). Since the chemical composition of PM25 changed significantly, it is necessary to study the relevant                                  |                                           |
|     | response of particle acidity based on high-resolution observation datasets.                                                                        |                                           |
|     | In this study, we calculated the bulk particle pH is calculated with the thermodynamic model ISORROPIA II in                                       | Formatted: Line spacing: 1.5 lines |
|     | forward mode with the assumption of aerosol in metastable state. The simulation was is limited to the data with the                                |                                           |
| 345 | corresponding RH between 20%-90%, same as that in previous studies (Liu et al. 2017; Cheng et al. 2016). The                                       |                                           |
|     | analysis is further limited to data when RH was between 20% 90%, same as previous studies (Liu et al.,                                             |                                           |
|     | 2017a;Cheng et al., 2016b). We further limit the analysis to data with sufficient aerosol liquid water (ALWC,                                      |                                           |
|     | (above $5\mu g/m^2 m^{-3}$ ) to avoid unrealistic pH values caused by false prediction predictions of ALWC. To study the                           |                                           |
| ĺ   | effect of the nitrate fraction's elevation on particle acidity, the Ratio N Ratio N-to-S wasis compared to the bulk particle |                                           |
|     | 15                                                                                                                                                 |                                           |

| 350 | pH (Fig6). As the nitrate fraction increased, increases, the particle pH also increased increases. When the ratio is                                                             |                                          |
|-----|----------------------------------------------------------------------------------------------------------------------------------------------------------------------------------|------------------------------------------|
|     | between 0-2, predicted pH value wasvalues are rather scattered (2.16.2) with a median value of 4.4. As the ratio                                                                 |                                          |
|     | increases, pH values becamebecome less scattered and the median value increased increases as well. When the ratio                                                                |                                          |
|     | wasis around 46, the predicted pH couldvalues range from 4.9 to 5.6 with a median value of 5.4, which is favorable                                                               |                                          |
|     | for aqueous oxidation of SO 2 by NO 2 (Cheng et al., 2016a;Wang et al., 2016a;Xie et al., 2015b). The comparable                                    |                                          |
| 355 | with previous reported values in PM 2.5 pollutions (Cheng et al. 2016; Wang et al. 2016; Xie et al. 2015). There are                                                  |                                          |
|     | several factors causing the chemistry nature of the pH increasing with higher Ratio N-to-S could be attributed to several                                             |                                          |
|     | reasons:, including neutralization by ammonia, higher pH of ammonium nitrate in comparison with ammonium                                                                         |                                          |
|     | sulfate, and increased ALWC ledleading to dilution of predicted H + (Hodas et al., 2014;Xue et al., 2014;Wang et al.,                                                 |                                          |
|     | 2018c). (Hodas et al. 2014; Xue et al. 2014; Wang et al. 2018c). To confirm that the pH elevation wasis not caused                                                               |                                          |
| 360 | by crustal ions, the simulation using data without crustal ions (input is set to zeroes 0) was conducted. It is shown                                                            |                                          |
|     | that the exclusion of crustal ions in the simulation could result can cause an overall lower pH, but the pH elevation                                                            |                                          |
|     | <del>aswith</del> Ratio N-to-S <del>wasis still observed. For details, please refer to (detailed analysis can be found in</del> the supplementary              |                                          |
|     | materials <del>(Fig., Figs.</del> S2- <del>Fig. and</del> S3).                                                                                                                   |                                          |
|     | Less In this study, fewer predicted H + ionions in aerosol liquid water waswere found to be the major cause of                                                        | Formatted: Left, Line spacing: 1.5 lines |
| 365 | the higher pH with high nitrate fraction in this study. The correlation between H + and major anions (HSO 4 - , SO 4 2- , |                                          |
|     | NO3 - , Cl - ) wasis shown in Fig.7 to identify the acidity contribution of each anion. Sulfate and bisulfate hadhave                                      |                                          |
|     | long been recognized as major acidacidic components of atmospheric particles. Their concentrations have                                                                          |                                          |
|     | significant impact on the particle acidity of particles (Weber et al., 2016;Liu et al., 2017a).(Weber et al. 2016; Liu                                                           |                                          |
|     | et al. 2017). Therefore, the H + ion concentration was found to strongly correlated correlate with sulfate as well as                                                 |                                          |
| 370 | bisulfate (Fig.Figs. 7a & Fig.and 7b). The outliers mightoutlier data points can be attributed to the fireworks' effect                                                          |                                          |

bisulfate (Fig. Figs. 7a & Fig. and 7b). The outliers mightoutlier data points can be attributed to the fireworks' eff events during spring festivalthe Chinese New Year (extreme data on Spring Festival's eve were Chinese New Year's Eve are excluded). The average molar ratio of bisulfate to sulfate is  $1.08 \times 10^{-6}$ , indicating that

|     | most of the sulfate wasis balanced by ammonium, same as previous studies (Song et al., 2018). the results reported                                        |
|-----|-----------------------------------------------------------------------------------------------------------------------------------------------------------|
|     | by previous studies (Song et al. 2018). The excess ammonium wasis then balanced by nitrate and chloride. The                                              |
| 375 | correlation between H + and nitrate ion wasis much different as ALWC varied varies (Fig. 7c). At Under the high-                               |
|     | ALWC condition, the H + increases as with the nitrate concentration increase, which can be explained by the                                    |
|     | simultaneously increased increasing sulfate fraction during several pollution episodes. At Under the drier condition                                      |
|     | (ALWC < $10\mu g/m^3 m^{-3}$ ), as NO 3 - increases, H + was decreasing decreases, which implies that the weaker aerosol |
|     | acidity favored favors nitric acid partitioning to the particle phase. Since HCl is more volatile than nitric acid gas,                                   |
| 380 | it'sits occurrence in the particle phase is more sensitive to the particle acidity of particles (Fig. 7d). Therefore, the                                 |
|     | negative correlation with H + wasis much obvious when it camecomes to chloride, free of ALWC amount.                                           |
|     | Discussions above showed that sulfate is the main particle acidity contributor in PM2.5 in Beijing. Given excess H+                                       |
|     | in the liquid water, HSO4 - would be formed as the equilibrium theory predicts. Therefore, HSO4 - to SO4 2- ratio could  |
|     | be a good indicator of aerosol's ability of excess H + formation. To understand the chemical nature of the elevation                           |
| 385 | of pH with increasing Ratio N to S , correlation between bisulfate/sulfate and Ratio N to S was investigated (Fig.8). The           |
|     | bisulfate/sulfate ratio significantly decreased when nitrate fraction increased, indicating that there were less free H+                                  |
|     | in ALWC when nitrate dominates the chemical composition of PM 2.5 . From these results and the fact that moderate                              |
|     | pH is more favorable for the partitioning of nitric acid gas to particle phase (Shah et al., 2018), the larger the fraction                               |
|     | of nitrate in PM2.5 is, the more balancing of anion by ammonium and less H + would be expected.                                                |
| 390 | The low level of H + , especially its negative correlation with Ratio N to S should be attributed to the neutralization by          |
|     | ammonia. Under most conditions, the excess of ammonia is an implicit prerequisite for SIA formation in Beijing,                                           |
|     | and the excessing level would affect the predicted particle acidity (Guo et al., 2017a; Weber et al., 2016). As an                                        |
|     | auxiliary evidence, the observed ammonia partition fraction (F NH4 ) was investigated to quantify the ammonia excess                           |
|     | and its relation with particle acidity. It could be easily seen that F NH4 exhibited a positive trend as Ratio N-0-S increases      |
| 395 | (Fig.9). The trend is divided into two branches colored by predicted pH: a more acidic (pH below 4.5) branch with                                         |
|     | Ratio-N-16-S range between 1 to 3 and F-NH4 ranges between 0.1 to 0.6; and another less acidic (pH above 5.5) branch                                      |
|     | with Ratio N to S range in 1 to 7 and F.NH4 range between 0.1 to 0.4. When airborne particles exhibit higher acidity, it                                  |
|     | is more favorable for ammonia partitioning to particle phase. Also, sulfate could accommodate twice the ammonia                                           |
|     | than nitrate, making the higher FNH4 in the upper branch. On the other hand, increased particle pH would prevent                                          |
| 400 | ammonia from partitioning to particle phase (Guo et al., 2017a), and the decrease in PM2.5-sulfate concentration                                          |
|     | might probably lead to higher ammonia concentration in the atmosphere according to recent study (Liu et al., 2018).                                       |

Therefore, the elevation of  $F_{NH4}$  at high pH (5~6) along with increasing RatioN-to-S (between 1 to 4) implied that the 17

nitrate formation was promoted by higher tropospheric ammonia concentration (Wang et al., 2013). When RatioN. to-5 was above 4, FNH4 decreased as pH increased. From above discussions, observation and model results showed 405 that fine particle enriched in nitrate during winter in Beijing will lead to lower particle acidity. The low level of H+, especially its negative correlation with RatioN-to-S should be attributed to the neutralization by ammonia via gas-particle partitioning. Under most conditions, the excess of ammonia is an implicit prerequisite for SIA formation in Beijing, and higher NH3 concentration could increase the predicted particle pH (Guo et al. 2017; Weber et al. 2016). As an auxiliary evidence, ammonia partition fraction (ENH4. 410 calculated with observation data) exhibits a positive trend as RatioN-to-S increases (Fig. 8), while ammonia concentration remains less varied in the same case (Fig.S4). The positive trend is divided into two parts: a more acidic (pH below 4.5) branch with RatioN-to-S of 1–3 and FNH4 of 0.1–0.6 and the other less acidic (pH above 5.5) branch with Ratio N-to-S of 1–7 and FNH4 of 0.1–0.4. The overall higher FNH4 in the lower pH branch is reasonable\_ since it is more favorable for ammonia partitioning to the particle phase when airborne particles exhibit higher 415 acidity. Moreover, sulfate can accommodate twice amount of ammonia than nitrate and thus increase  $F_{NH4}$ . Yet, the highest value of  $F_{NH4}$  were observed with more nitrate (Ratio N-40-S ~2.5) and a slightly higher pH. By contrast, even though the high particle pH (5~6) can prevent ammonia from partitioning to the particle phase (Guo et al. 2017), the elevation of FNH4 with increasing RatioN-to-S (1–4) is still observed with pH ranging 5 to 6 despite there were some outliers with lower  $F_{NH4}$  which is not caused by higher  $NH_3$  concentration. Nitrate formation is 420 observed to be enhanced in North China, either by heterogeneous formation (e.g. N2O5 hydrolysis) or with sufficient ambient NH3 (Wang et al. 2013). The positive trend of FNH4 with RatioN-to-S clearly shows that nitrate formation and partitioning have significant contribution to NH3 partitioning process, and will lead to an enhanced neutralization with the help of more ammonia partitioning into the particle phase. Combining these analyses, we conclude that fine particles enriched in nitrate in Beijing during winter will lead to lower particle acidity.

| 425 | 4. Discussion: Possible possible impacts of increasing fraction of nitrate in PM2.5                                                   | Fe               | ormatted: Line spacing: 1.5 lines                           |
|-----|---------------------------------------------------------------------------------------------------------------------------------------|------------------|-------------------------------------------------------------|
|     |                                                                                                                                       |                  |                                                             |
|     | So far, the effect of emission control's effect on SNA composition compositions in Beijing's PM 2.5 pollution              | Fo               | ormatted: Left, Indent: First line: 2 ch, Line spacing: 1.5 |
|     | and the response of particle pH washave been illustrated, but it is important to make future predictions with the                     |                  |                                                             |
|     | currentlycurrent knowledge for future better control strategy. In this section, sensitivity tests regarding to                        |                  |                                                             |
|     | hygroscopicity and particle-acidity change wereare conducted to help understand the possible changes of these                         |                  |                                                             |
| 430 | properties in the future. Aerosol liquid water content (ALWC) wasALWC is directly engaged in the calculation of                       |                  |                                                             |
|     | particle pH and limited by several major parameters (relative humidity RH , hydrophilic composition                            |                  |                                                             |
|     | concentrations, temperaturesconcentration, temperature, etc.). During the campaign, the ALWC predicted by                             |                  |                                                             |
|     | ISORROPIA II varied between 0.8 to and 154.2 $\mu g/m^3 m^{-3}$ with an average value of 6.4 $\mu g/m^3 m^{-3}$ . As previously       |                  |                                                             |
|     | mentioned, the average value of Ratio N-to-S wasis around 2 during the haze eventevents in the winter of 2017 in           |                  |                                                             |
| 435 | Beijing. The average ALWC in the haze events increased to $24.4 \mu g/m^3  \underline{m^{-3}}$ accordingly. It has been reported that |                  |                                                             |
|     | nitrate salts have largergreater contribution to ALWC due to its lower deliquescence RH, and the elevated ALWC                        |                  |                                                             |
|     | might have has strong impact on several water-soluble gas partitioning such as glyoxal, leading to further secondary                  |                  |                                                             |
|     | composition formation processes (Hodas et al., 2014;Xue et al., 2014).an enhanced SOA production (Hodas et al.                        |                  |                                                             |
|     | 2014; Xue et al. 2014). As a matter of fact, the increase of hygroscopicity related to nitrate-rich fine particles                    |                  |                                                             |
| 440 | havehas been observed in Beijing (Wang et al., 2018c). (Wang et al. 2018c). However, though the possibly higher-                      |                  |                                                             |
|     | ALWC in nitrate rich particles might lead to dilution of H + , it was it is difficult to conclude that the lower pH in     |                  |                                                             |
|     | nitrate-rich particles have is caused by the dilution of H + with higher ALWC with current data, since the higher          | F c       | ormatted: Superscript                                       |
|     | nitrate fraction wasis usually observed with the moderate RH in pollution episodes in the winter of 2017.                             | Fo               | ormatted: Left, Indent: First line: 0.59 cm, Line spacing:  |
|     | Moving on now to consider the The possible enhancement of hygroscopicity in nitrate_rich PM25. For most                               | 1.:
Fo        | 5 lines prmatted: Font color: Black                         |
| 445 | was investigated. Most single salts, can only be deliquesced over a certain RH-it can be deliquesced, which often                     | F                | ormatted: Font color: Black                                 |
|     |                                                                                                                                       | F                | ormatted: Font color: Black                                 |
|     | behaves in a way that, thus the ALWC only exists when a certain RH is exceeded (Wexler and Seinfeld,                                  | - ( ) ( F | ormatted: Font color: Black                                 |
|     | 1991;Mauer and Taylor, 2010). However, the atmospheric(Wexler and Seinfeld 1991; Mauer and Taylor 2010). In                           | F                | ormatted: Font color: Black                                 |
| 1   | 10                                                                                                                                    | Fo               | ormatted: Font color: Black                                 |
|     | 19                                                                                                                                    |

---

## Author Response (AR2)

**Dear ACP Editor, Dear Reviewers**

Thank you very much for taking your time to review this manuscript. We really appreciate all your comments and suggestions! After carefully reading the comments from you, we have revised our manuscript. Your suggestions have enabled us to improve our work. Please find our itemized responses in below and our revisions/corrections in the resubmitted files. Accordingly, we have uploaded a copy of the original manuscript with all the changes highlighted by using the track changes mode in MS Word.

We would like also to thank you for allowing us to resubmit a revised copy of the manuscript. We hope that the revised manuscript is accepted for publication in Atmospheric Chemistry and Physics.

Sincerely,

Yuning Xie & Gehui Wang

**Editor:**

**Comments to the authors:**
*"In addition to copy editing reqired by Referee #1, Referee #3 commented on one key finding and asks from additional sensitivity tests. You are invited to revise the manuscript and prepared detailed responses."*

**Authors' response to editor:**
These suggestions are all taken carefully. Based on the referees' comments, the major revision in the prepared paper is the additional content of the suggested sensitivity test and presentation on strengthening the main conclusions. We added the results of pH response when transferring sulfate to nitrate and some other supportive materials. Then, the copy editing was conducted throughout the manuscript. We hope these works to be informative and helpful for this study.

**Referee #2:**

**Comments to the authors:**
*"Copy editing required."*

**Authors' response to Referee #2:**
We have carefully corrected all the typos. We may include all text editing in the response to Referee #2. Here are examples of the corrections (row numbers refer to the revised manuscript), detail modifications could be found in the comparison MS document:

1.  Page 1, line 15: "Strict pollution control actions also reduced the average $PM_{2.5}$ concentration to a low level of 39.7 $\mu g\, m^{-3}$ in urban Beijing during the winter of 2017"

moved to line 15 to make the introduction more logic.

2. Page 1, Line 17: Correction of wrong expressions. "chemistry of" modified to "physiochemical properties".

3. Page 2, Line 25: To make the abstract more comprehensive, a sentence introducing the results from suggested sensitivity tests - "Controlled variable tests showed that the pH elevation should be attributed to nitrate fraction increase other than crustal ion and ammonia concentration increases."

4. Page 2, Line 39: Informal language use corrected - "$PM_{2.5}$ pollution is relieved …" modified to "$PM_{2.5}$ pollution occurrence is reduced to …".

5. Page 3, Line 52 - 53: Break long sentences into shorter sentences to increase readability. "It was reported …by $H_2O_2$, but recent …" modified to "It was reported … by $H_2O_2$. But recent …". Deleted the word "$H_2O_2$" and "which are mentioned in the textbooks" in the sentence "But recent studies …".

6. Page 3, Line 65: Author's name in reference changed according to accurate document. "Robert et al. 2016" changed to "Oleniaczs et al. 2016", and the authors' names in Reference were also changed.

7. Page 3, Line 67: Overlap in content modified – remove "which can calculate the particle pH".

8. Page 7, Line 155: Reduction of unnecessary words – deleted "major secondary" and rephrased the next sentence.

9. Page 7 – 8, Line 160 – 165: Combination of above-mentioned language corrections to make the presentation more readable.

10. Page 8, Line 175 – 176: Adding of precise molar concentration for $K^+$ and $Cl^-$ for better presentation.

11. Page, 10, Line 211: Remove vague sentences to avoid misunderstanding – remove "The decreasing trend of ammonium, which is the major neutralizer in the atmosphere of Beijing, well represented the efficient pollution control of SNA compositions during the winter of 2017".

There are more copy editing throughout the manuscript. We kindly invite the editor and both referees to read the revised version. For now, it is our best effort to improve the presentation quality.

**Referee #3:**

**Comments to the authors:**

*"As mentioned in its title and abstract, the key argument made in this paper is that 'As nitrate's fraction significantly elevated, particle pH was also found to increase in winter Beijing given sufficient ammonia (average concentration 7.1μg/m3, 12.9μg/m3 during pollution). During PM2.5 pollution episodes, the particle pH predicted increased*

*from 4.4 (moderate acidic) to 5.4 (near neutral) as nitrate to sulfate molar ratio increased from 1 to 5.'*

*Currently, this argument is mainly supported by Figure 6 "Scatter plot of simulated pH vs. the molar ratio of nitrate to sulfate", when only the data during pollution classification and with sufficient aerosol liquid water (above 5μg/m3) was chosen. The authors also use sensitivity studies to exclude the potential effect of crustal elements on the results.*

*As we all know, correlation does not necessarily imply causation. I suggest the authors make additional sensitivity tests to rule out other possible reasons for the observed correlation between pH and Ratio(N-to-S). The possibilities include: (1) ammonia. Will the authors find a relationship between gaseous $NH_3$ and pH? Previous literature, some of which are citied here, show the positive relationship between pH and $log10(NH_3)$; (2) ambient relative humidity RH. Will the authors find a relationship between pH and RH? Higher RH may lead to higher aerosol water amount and then higher pH. (3) Temperature. Higher temperature may lead to lower pH. These sensitivity tests should be conducted before drawing the conclusion that elevated nitrate fraction leads to elevated pH.*

*In fact, the best way to demonstrating the relationship between nitrate fraction and pH, I suggest, is to replace nitrate with the same moles of sulfate in the input of the thermodynamic equilibrium model, or to replace sulfate with the same moles of nitrate in the input. In this way, all the other model inputs, including RH, temperature, and other cations/anions, are kept the same and assumed as control variables. I highly recommend the authors conduct such sensitivity tests.*

*These suggested sensitivity analyses imply a major revision to the current paper."*

**Authors' response to Referee #3**:

Suggestion taken. We have added more content to address the suggestions given by referee#3. As suggested by the referee, in order to demonstrate our conclusions on the relationship between nitrate fraction and pH, we did additional sensitivity test by replacing sulfate with the same moles of nitrate in the input, and keeping other parameters constantly, including RH, temperature, ammonia and other cations and anions. As seen in Figure 7 in the manuscript and below, the sensitivity test results showed a continuous increase in pH when sulfate was step wisely replaced by nitrate, again demonstrating our argument that as nitrate's fraction significantly elevated, particle pH was also found to increase in winter Beijing. To be clearer and in accordance with the reviewer concerns, we have added a brief description as follows (also could be found on page 12, line 254 – 265 in the revised manuscript):

*"In order to further elucidate the relationship between nitrate fraction and pH, a controlled sensitivity test was conducted for the Beijing PM 2.5 aerosols by replacing particulate sulfate with the same moles of nitrate and keeping all other variables constant such as RH, temperature, ammonia, anions and cations. As shown in Fig.7, the median values of simulated pH increased from 4.6 to 5.1 as the transferring fraction increases from 10% to 80%. When the fraction exceeded 80%, pH median value was a bit lower compare to the pH80% (Fig.7.a). By examining the difference between the sensitivity test results and the original pH values, an overall increase of pH by ~0.5 was found when the*

*fraction exceeds 60% (Fig.7.b). The detail sensitivity results and description also showed that the pH elevation due to the replacement of sulfate by nitrate was observed at all conditions (Fig.S5). With the fact that other variables remained controlled, the test results reconfirm that as nitrate become dominant, particle pH would significantly increase."*

Besides the suggestion on the above-mentioned test, the referee kindly suggest that relative humidity, temperature and ammonia could have impact on the pH elevation. Our analysis suggest that these are poorly correlated with the molar ratio of nitrate to sulfate (Fig.S4, Fig.S6, Fig.S7). These three parameters have different impact on pH: (1) higher relative humidity would led to higher ALWC and might dilute $H^+$, thus increase the calculated pH; (2) Temperature will affect pH, but the difference is usually less noticeable; (3) Higher ammonia concentration will effectively increase particle pH from previous studies. Yet, the poor correlation proves that when nitrate fraction increases, the RH (depict as ALWC), temperature as well as ammonia do not change as they ought to be. In conclusion, these variables are not the cause of pH elevation in this study. Short supplementary content was added to the manuscript as follows (also could be found on page 14, line 315 – 319 in the revised manuscript):

*"Furthermore, the elevation of pH due to $Ratio_{N\text{-}to\text{-}S}$ increase is one unit of pH, which means the ALWC shall increase 10 times of amount. This assumption is not supported by current data, since ALWC remain less varied as $Ratio_{N\text{-}to\text{-}S}$ increases (Fig.S6). Similar with ALWC, the correlation between ambient temperature and $Ratio_{N\text{-}to\text{-}S}$ is low, further proving that the increase of pH is not caused by change of thermodynamic state (Fig.S7)."*

Once again, we appreciated the referee for his/her kind suggestions. These additional analyses provide the in-depth investigation on an observation-based particle pH and its link with chemical compositions during winter in Beijing. We hope these analyses could provide enough supplementary evidence for our conclusions.

**Results of the suggested sensitivity test**

[Figure]

**Fig.7** Box plot of **(a)** simulated pH using the setting of the sensitivity test as well as **(b)** pH difference between the simulated pH from the transferred data and the original observation data (Only data with

sufficient ALWC during the pollution period were shown).

**Correlation between ammonia, ALWC, Temperature and Ratio$_{N-to-S}$**

[revised manuscript text omitted]